# Tailored design of protein nanoparticle scaffolds for multivalent presentation of viral glycoprotein antigens

George Ueda[1,2†], Aleksandar Antanasijevic[3,4†], Jorge A Fallas[1,2], William Sheffler[1,2], Jeffrey Copps[3,4], Daniel Ellis[1,2], Geoffrey B Hutchinson[5], Adam Moyer[1,2], Anila Yasmeen[6], Yaroslav Tsybovsky[7], Young-Jun Park[1], Matthew J Bick[1,2], Banumathi Sankaran[8], Rebecca A Gillespie[5], Philip JM Brouwer[9], Peter H Zwart[8,10], David Veesler[1], Masaru Kanekiyo[5], Barney S Graham[5], Rogier W Sanders[6,9], John P Moore[6], Per Johan Klasse[6], Andrew B Ward[3,4*], Neil P King[1,2*], David Baker[1,2,11*]

[1]Department of Biochemistry, University of Washington, Seattle, United States; [2]Institute for Protein Design, University of Washington, Seattle, United States; [3]Department of Integrative Structural and Computational Biology, The Scripps Research Institute, La Jolla, United States; [4]International AIDS Vaccine Initiative Neutralizing Antibody Center, the Collaboration for AIDS Vaccine Discovery (CAVD) and Scripps Consortium for HIV/AIDS Vaccine Development (CHAVD), The Scripps Research Institute, La Jolla, United States; [5]Vaccine Research Center, National Institute of Allergy and Infectious Diseases, National Institutes of Health, Bethesda, United States; [6]Department of Microbiology and Immunology, Weill Cornell Medicine, Cornell University, New York, United States; [7]Electron Microscopy Laboratory, Cancer Research Technology Program, Frederick National Laboratory for Cancer Research sponsored by the National Cancer Institute, Frederick, United States; [8]Berkeley Center for Structural Biology, Molecular Biophysics and Integrated Bioimaging, Lawrence Berkeley Laboratory, Berkeley, United States; [9]Amsterdam UMC, Department of Medical Microbiology, Amsterdam Infection & Immunity Institute, University of Amsterdam, Amsterdam, Netherlands; [10]Center for Advanced Mathematics in Energy Research Applications, Computational Research Division, Lawrence Berkeley Laboratory, Berkeley, United States; [11]Howard Hughes Medical Institute, University of Washington, Seattle, United States

**\*For correspondence:**
andrew@scripps.edu (ABW);
neilpking@gmail.com (NPK);
dabaker@uw.edu (DB)

†These authors contributed equally to this work

**Abstract** Multivalent presentation of viral glycoproteins can substantially increase the elicitation of antigen-specific antibodies. To enable a new generation of anti-viral vaccines, we designed self-assembling protein nanoparticles with geometries tailored to present the ectodomains of influenza, HIV, and RSV viral glycoprotein trimers. We first *de novo* designed trimers tailored for antigen fusion, featuring N-terminal helices positioned to match the C termini of the viral glycoproteins. Trimers that experimentally adopted their designed configurations were incorporated as components of tetrahedral, octahedral, and icosahedral nanoparticles, which were characterized by cryo-electron microscopy and assessed for their ability to present viral glycoproteins. Electron microscopy and antibody binding experiments demonstrated that the designed nanoparticles presented antigenically intact prefusion HIV-1 Env, influenza hemagglutinin, and RSV F trimers in the predicted geometries. This work demonstrates that antigen-displaying protein nanoparticles can be designed from scratch, and provides a systematic way to investigate the influence of antigen presentation geometry on the immune response to vaccination.

## Introduction

Multivalent antigen presentation, in which antigens are presented to the immune system in a repetitive array, has been demonstrated to increase the potency of humoral immune responses (*Bennett et al., 2015*; *Snapper, 2018*). This has been attributed to increased cross-linking of antigen-specific B cell receptors at the cell surface and modulation of immunogen trafficking to and within lymph nodes (*Irvine et al., 2013*; *Tokatlian et al., 2019*). An ongoing challenge has been to develop multimerization scaffolds capable of presenting complex oligomeric or engineered antigens (*Sanders and Moore, 2017*; *Jardine et al., 2013*; *McLellan et al., 2013a*), as these can be difficult to stably incorporate into non-protein-based nanomaterials (e.g. liposomes, polymers, transition metals and their oxides). Epitope accessibility, proper folding of the antigen, and stability are also important considerations in any strategy for antigen presentation. Several reports have utilized non-viral, naturally occurring protein scaffolds, such as self-assembling ferritin (*Kanekiyo et al., 2013*; *Sliepen et al., 2015*; *Darricarrère et al., 2018*), lumazine synthase (*Sanders and Moore, 2017*; *Abbott et al., 2018*), or encapsulin (*Kanekiyo et al., 2015*) nanoparticles, to present a variety of complex oligomeric or engineered antigens. These studies have illustrated the advantages of using self-assembling proteins as scaffolds for antigen presentation (*López-Sagaseta et al., 2016*; *Kanekiyo et al., 2019*), including enhanced immunogenicity and seamless integration of antigen and scaffold through genetic fusion. More recently, computationally designed one- and two-component protein nanoparticles (*Hsia et al., 2016*; *King et al., 2014*; *Bale et al., 2016*) have been used to present complex oligomeric antigens, conferring additional advantages such as high stability, robust assembly, ease of production and purification, and increased potency upon immunization (*Marcandalli et al., 2019*; *Brouwer et al., 2019*).

The ability to predictively explore new structural space makes designed proteins (*Parmeggiani et al., 2015*; *Brunette et al., 2015*) attractive scaffolds for multivalent antigen presentation. In our previous work with computationally designed nanoparticle immunogens (*Marcandalli et al., 2019*; *Brouwer et al., 2019*), the nanoparticles were generated from naturally occurring oligomeric proteins without initial consideration of geometric compatibility for antigen presentation. A more comprehensive solution would be to *de novo* design nanoparticles which present complex antigens of interest. For homo-oligomeric class I viral fusion proteins, a large group that includes many important vaccine antigens (*Harrison, 2015*), a close geometric match between the C termini of the antigen and the N termini of a designed nanoparticle component would enable multivalent presentation without structural distortion near the glycoprotein base, and potentially allow for better retention of antigenic epitopes relevant to protection. More generally, precise control of antigen presentation geometry through *de novo* nanoparticle design would enable systematic investigation of the structural determinants of immunogenicity.

## Results

### *De novo* design of protein nanoparticles tailored for multivalent antigen presentation

We sought to develop a general computational method for *de novo* designing protein nanoparticles with geometries tailored to present antigens of interest, focusing specifically on the prefusion conformations of the trimeric viral glycoproteins HIV-1 Env (BG505 SOSIP) (*Wang et al., 2017*; *Sanders et al., 2013*), influenza hemagglutinin (H1 HA) (*Kadam et al., 2017*), and respiratory syncytial virus (RSV) F (DS-Cav1) (*McLellan et al., 2013a*). To make the antigen-tailored nanoparticle design problem computationally tractable, we employed a two-step design approach (*Figure 1*). In the first step, we *de novo* designed antigen-tailored trimers, featuring N termini geometrically matched to the C termini of the viral glycoproteins. In the second step, we generated tetrahedral, octahedral, and icosahedral two-component nanoparticles by designing secondary interfaces between a designed trimer (fusion component) and a *de novo* homo-oligomer (assembly component) (*Fallas et al., 2017*). This design approach yielded nanoparticles tailored to present 4, 8, or 20 copies of the viral glycoproteins in defined geometries (*Figure 1d*). Sequences for all designed trimers and homo-oligomers, two-component nanoparticles, and antigen-fused components in this study can be found in *Supplementary file 1A, B, and C*, respectively. Details on each step of the design approach are described in the following sections.

**eLife digest** Vaccines train the immune system to recognize a specific virus or bacterium so that the body can be better prepared against these harmful agents. To do so, many vaccines contain viral molecules called glycoproteins, which are specific to each type of virus. Glycoproteins that sit at the surface of the virus can act as 'keys' that recognize and unlock the cells of certain organisms, leading to viral infection.

To ensure a stronger immune response, glycoproteins in vaccines are often arranged on a protein scaffold which can mimic the shape of the virus of interest and trigger a strong immune response. Many scaffolds, however, are currently made from natural proteins which cannot always display viral glycoproteins.

Here, Ueda, Antanasijevic et al. developed a method that allows for the design of artificial proteins which can serve as scaffold for viral glycoproteins. This approach was tested using three viruses: influenza, HIV, and RSV – a virus responsible for bronchiolitis. The experiments showed that in each case, the relevant viral glycoproteins could attach themselves to the scaffold. These structures could then assemble themselves into vaccine particles with predicted geometrical shapes, which mimicked the virus and maximized the response from the immune system.

Designing artificial scaffold for viral glycoproteins gives greater control over vaccine design, allowing scientists to manipulate the shape of vaccine particles and test the impact on the immune response. Ultimately, the approach developed by Ueda, Antanasijevic et al. could lead to vaccines that are more efficient and protective, including against viruses for which there is currently no suitable scaffold.

## Computational design of trimers tailored for fusion to specific viral glycoproteins

We chose to design our antigen-tailored trimers from monomeric repeat proteins composed of rigidly packed 20- to 50-residue tandem repeat units (*Parmeggiani et al., 2015*; *Brunette et al., 2015*; *Urvoas et al., 2010*; *Kajander et al., 2007*; *Main et al., 2003*), as their high stability and tunable length (through variation of repeat number) are desirable properties for the design of protein-based nanomaterials. These structurally diverse alpha-helical repeat proteins featured three to six repeat modules and total lengths between 119 and 279 residues. They were docked into C3-symmetric trimers using our RPX docking method, which identifies configurations likely to accommodate favorable side chain packing at the *de novo* designed interface (*Fallas et al., 2017*). To identify trimeric configurations with N termini compatible for fusion to the C termini of the three viral glycoproteins, docks with an RPX score above 5.0 were screened using the sic_axle protocol (*Marcandalli et al., 2019*). Geometrically compatible docks (non-clashing termini separation distances of 15 Å or less) were subjected to full Rosetta C3-symmetric interface design and filtering (see Materials and Methods), and twenty-three designs were selected for experimental characterization (*Figure 1—figure supplement 1*).

## Structural characterization of designed trimers

Synthetic genes encoding each of the designed trimers were expressed in *E. coli* and purified from lysates by $Ni^{2+}$ immobilized metal affinity chromatography ($Ni^{2+}$ IMAC) followed by size-exclusion chromatography (SEC). Twenty-two designs were found to express in the soluble fraction, and nine formed the intended trimeric oligomerization state as assessed by SEC in tandem with multi-angle light scattering (SEC-MALS; examples in *Figure 2* top panel, second row; SEC-MALS chromatograms for the remaining designs are in *Figure 2—figure supplement 2* and data in *Figure 2—figure supplement 1—source data 1*; SEC chromatograms for remaining designs with off-target retention volumes are in *Figure 2—figure supplement 2*). Four of the designs that were trimeric and expressed in high yield, 1na0C3_2, 3ltjC3_1v2, 3ltjC3_11, and HR04C3_5v2, were selected for solution small angle X-ray scattering (SAXS) experiments. The proteins exhibited scattering profiles very similar to those computed from the corresponding design models, suggesting similar supramolecular configuration (*Figure 2* top panel, third row; metrics in *Table 1* and *Figure 2—source data 1*). These four trimers were derived from three distinct designed helical repeat proteins from TPR,

**Figure 1.** *De novo* design of protein nanoparticles tailored for multivalent antigen presentation. (**a**) Computational docking of monomeric repeat proteins into C3-symmetric trimers using the RPX method. (**b**) Selection of trimers for design based on close geometric match between their N termini (blue spheres) and C termini (red spheres) of a viral antigen (green, BG505 SOSIP shown for illustration). (**c**) Design of two-component nanoparticles incorporating a fusion component (cyan) and assembly component (gray). (**d**) Nanoparticle assembled with antigen-fused trimeric component yields multivalent antigen-displaying nanoparticle.

The online version of this article includes the following figure supplement(s) for figure 1:

**Figure supplement 1.** Computational docking and design of trimers for fusion to a specific viral glycoprotein.

HEAT, or *de novo* topological families (1na0, 3ltj, and HR04, see Materials and Methods) (*Brunette et al., 2015*; *Urvoas et al., 2010*; *Main et al., 2003*). Crystals were obtained for the two designs 1na0C3_2 and 3ltjC3_1v2. Structures were determined at resolutions of 2.6 and 2.3 Å, revealing a backbone root mean square deviation (r.m.s.d.) between the design model and structure of 1.4 and 0.8 Å, respectively (*Figure 2—figure supplement 3*, and *Figure 2—figure supplement 3—source data 1*, crystallization conditions, structure metrics, and structure-to-model comparisons are described in Materials and Methods). The structures confirmed in both cases that the designed proteins adopt the intended trimeric configurations, and that most of the atomic details at the *de novo* designed interfaces are recapitulated.

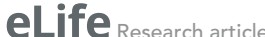

**Figure 2.** Biophysical characterization of antigen-tailored trimers and nanoparticles. Top rows, design models. Middle rows, SEC chromatograms and calculated molecular weights from SEC-MALS. Bottom rows, comparisons between experimental SAXS data and scattering profiles calculated from design models. (**a**) 1na0C3_2. (**b**) 3ltjC3_1v2. (**c**) 3ltjC3_11. (**d**) HR04C3_5v2. (**e**) T33_dn2. (**f**) T33_dn10. (**g**) O43_dn18. (**h**) I53_dn5.

*Figure 2 continued on next page*

*Figure 2 continued*

The online version of this article includes the following source data and figure supplement(s) for figure 2:

**Source data 1.** Biophysical properties of designed trimers and two-component nanoparticles.
**Source data 2.** 1na0C3_2 SEC-MALS.
**Source data 3.** 3ltjC3_1v2 SEC-MALS.
**Source data 4.** 3ltjC3_11 SEC-MALS.
**Source data 5.** HR04C3_5v2 SEC-MALS.
**Source data 6.** 1na0C3_2 SAXS.
**Source data 7.** 3ltjC3_1v2 SAXS.
**Source data 8.** 3ltjC3_11 SAXS.
**Source data 9.** HR04_5v2 SAXS.
**Source data 10.** T33_dn2 SEC-MALS.
**Source data 11.** T33_dn10 SEC-MALS.
**Source data 12.** O43_dn18 SEC-MALS.
**Source data 13.** I53_dn5 SEC-MALS.
**Source data 14.** T33_dn2 SAXS.
**Source data 15.** T33_dn10 SAXS.
**Source data 16.** O43_dn18 SAXS.
**Source data 17.** I53_dn5 SAXS.
**Figure supplement 1.** SEC-MALS chromatograms for designed trimers occupying an off-target oligomeric state.
**Figure supplement 1—source data 1.** SEC-MALS data for off-target designed trimers.
**Figure supplement 2.** SEC chromatograms for designed trimers with off-target retention volumes after $Ni^{2+}$ IMAC.
**Figure supplement 3.** Comparison between the experimentally determined crystal structures and corresponding models of two designed trimers.
**Figure supplement 3—source data 1.** Crystallography data collection and refinement statistics for designed trimers 1na0C3_2 and 3ltjC3_1v2.
**Figure supplement 4.** SDS-PAGE of bicistronically-expressed designed nanoparticles eluted from $Ni^{2+}$ IMAC.
**Figure supplement 5.** SEC profiles for two-component nanoparticles with off-target retention volumes after $Ni^{2+}$ IMAC.
**Figure supplement 6.** Biophysical characterization of T33_dn5.
**Figure supplement 6—source data 1.** T33_dn5 SEC-MALS.
**Figure supplement 6—source data 2.** T33_dn5 SAXS.
**Figure supplement 7.** *In vitro* assembly of I53_dn5.

## Computational design of two-component nanoparticles incorporating designed trimers

As secondary assembly components were required to design our antigen-tailored nanoparticles, validated trimers were docked pairwise with *de novo* designed symmetric homo-oligomers (*Fallas et al., 2017*) to generate tetrahedral, octahedral, and icosahedral nanoparticle configurations using the TCdock program (*King et al., 2014*; *Bale et al., 2016*). To increase the probability of generating icosahedra which confer the highest valency among the targeted symmetries, three naturally occurring homopentamers were also included in the docking calculations (PDB IDs 2JFB, 2OBX, and 2B98). Analogously to the designed trimers, nanoparticle docks were scored and ranked using the RPX method (*Fallas et al., 2017*) to identify configurations likely to accommodate favorable side chain packing at a secondary *de novo* designed interface. High-ranking and non-redundant nanoparticle configurations featuring outward-facing N termini for antigen presentation were selected for Rosetta interface design (*King et al., 2014*; *Bale et al., 2016*). Fifty-three nanoparticle designs across all three targeted symmetries that exhibited the best interface metrics were selected for experimental characterization (see Materials and Methods). The nomenclature for the eleven tetrahedra, twenty-one octahedra, and twenty-one icosahedra indicate the symmetry of the nanoparticle (T, O, or I), the oligomeric state of the first component (A) and second component (B) used in each design, the letters "dn" reflecting the *de novo* nature of the input oligomers, and the rank by RPX score from the docking stage (e.g., "I53_dn5" indicates an icosahedral nanoparticle constructed from a pentameric and trimeric component, ranked 5th in RPX-scoring for the two input oligomers).

Synthetic genes encoding each of the two-component nanoparticles were obtained with one of the components fused to a $His_6$-tag, and the designs were purified using $Ni^{2+}$ IMAC (see Materials

**Table 1.** Summary of the experimental characterization for designed trimers and two-component nanoparticles. 1na0C3_2 and 3ltjC3_1v2 structures determined by X-ray crystallography and T33_dn10, O43_dn18, and I53_dn5 structures determined by cryo-EM.

| Design | Targeted Antigens | Experimental Molecular Weight (kDa) | Target Molecular Weight (kDa) | SAXS $X$ value | Resolution, backbone r.m.s.d. structure (Å, Å) |
|---|---|---|---|---|---|
| 1na0C3_2 | HA, SOSIP, DS-Cav1 | 48 | 45 | 1.4 | 2.6, 1.4 |
| 3ltjC3_1v2 | SOSIP, DS-Cav1 | 56 | 63 | 1.1 | 2.3, 0.8 |
| 3ltjC3_11 | SOSIP, DS-Cav1 | 50 | 66 | 1.6 | – |
| HR04C3_5v2 | SOSIP | 71 | 69 | 1.5 | – |
| T33_dn2 | HA, SOSIP, DS-Cav1 | 397 | 345 | 4.8 | – |
| T33_dn5 | HA, SOSIP, DS-Cav1 | 422 | 422 | 1.7 | – |
| T33_dn10 | HA, SOSIP, DS-Cav1 | 546 | 556 | 2.3 | 3.9, 0.65 |
| O43_dn18 | HA,SOSIP, DS-Cav1 | 810 | 876 | 2.9 | 4.5, 0.98 |
| I53_dn5 | HA, SOSIP, DS-Cav1 | 2000 | 1960 | 1.2 | 5.3, 1.30 |

The online version of this article includes the following source data for Table 1:

Source data 1. Summary of the experimental characterization for designed trimers and two-component nanoparticles.

and methods). Pairs of proteins at the expected molecular weights were found to co-elute by SDS-PAGE for twenty-four of the designs, consistent with spontaneous assembly of the nanoparticles followed by pulldown His$_6$-tagged component (featured co-eluting designs are presented in *Figure 2—figure supplement 4*). SEC chromatograms revealed that nineteen designs did not form assemblies of the expected size or that the resulting assemblies were heterogeneous (*Figure 2—figure supplement 5*). Five designs comprising a panel of unique geometric configurations, T33_dn2, T33_dn5, T33_dn10, O43_dn18, and I53_dn5, ran as monodisperse particles of the predicted molecular mass by SEC-MALS and were further investigated by SAXS. The experimental solution scattering curves closely matched the scattering curves computed from the design models (*Schneidman-Duhovny et al., 2010*) for all five designs (*Figure 2*, bottom panel and *Figure 2—figure supplement 6*; metrics in *Table 1* and *Figure 2—source data 1*, bottom five designs).

Due to its high valency and production yield, we selected the I53_dn5 nanoparticle to investigate the capacity of its two components to be separately produced and assembled *in vitro*. The two components of I53_dn5 were re-cloned, expressed, and separately purified (pentameric "I53_dn5A" with His$_6$-tag and trimeric "I53_dn5B"). Nanoparticle assembly appeared to be complete within minutes after equimolar mixing (*Figure 2—figure supplement 7*). This capability is noteworthy as it enables production of each component independently, even from different host systems, which provides more flexibility in nanoparticle manufacturing. *In vitro* assembly also confers more control over nanoparticle assembly and composition, for example by assembling with a mixture of components fused to different antigens (*Boyoglu-Barnum et al., 2020*).

## Structural characterization of designed two-component nanoparticles

The five SAXS-validated nanoparticles were structurally characterized using negative stain electron microscopy (NS-EM) (*Lee and Gui, 2016*; *Ozorowski et al., 2018*). 2,000–5000 particles were manually picked from the electron micrographs acquired for each designed nanoparticle and classified in 2D using the Iterative MSA/MRA algorithm (see Materials and Methods). 3D classification and refinement steps were performed in Relion/3.0 (*Zivanov et al., 2018*). Analysis of the NS-EM data confirmed high sample homogeneity for all five nanoparticle designs as evident from the micrographs and 2D class-averages (*Figure 3*). While some free nanoparticle components were detected in the T33_dn5 sample, suggesting a certain propensity towards disassembly, analysis of the reconstructed 3D maps revealed that all five nanoparticles assemble as predicted by the design models, at least to the resolution limits of NS-EM.

In order to obtain higher-resolution information, three designs, T33_dn10, O43_dn18, and I53_dn5, representing one nanoparticle from each targeted symmetry (T, O, I), were subjected to cryo-electron microscopy (cryo-EM). Cryo-EM data acquisition was performed as described in the Materials and Methods section and data acquisition statistics are displayed in *Figure 4—source*

**Figure 3.** NS-EM analysis of antigen-tailored nanoparticles. From left to right: designed trimers incorporated in each designed nanoparticle, nanoparticle design models fit into NS-EM density (views shown down each component axis of symmetry), designed nanoparticle 2D class-averages, raw electron micrographs of designed nanoparticles. (**a**) T33_dn2. (**b**) T33_dn5. (**c**) T33_dn10. (**d**) O43_dn18. (**e**) I53_dn5.

*data 1*. The data processing workflow is presented in *Figure 4—figure supplement 1*. Appropriate symmetry (T, O, and I for T33_dn10, O43_dn18, and I53_dn5, respectively) was applied during 3D classification and refinement and maps were post-processed in Relion/3.0 (*Zivanov et al., 2018*). The final resolutions of the reconstructed maps for the T33_dn10, O43_dn18, and I53_dn5 nanoparticles were 3.9, 4.5, and 5.3 Å, respectively. Some structural heterogeneity was observed in the cryo-EM data, particularly in the case of I53_dn5. In 2D classification results we generated particle projection averages that range from spherical to ellipsoid shape (*Figure 4—figure supplement 1c*), indicating some degree of flexibility. There is less evidence of flexibility in T33_dn10 and O43_dn18, in agreement with the higher final map resolution for these nanoparticles.

Nanoparticle design models were relaxed into the corresponding EM maps by applying multiple rounds of Rosetta relaxed refinement (*Wang et al., 2016*) and manual refinement in Coot (*Emsley and Crispin, 2018*) to generate the final structures. Refined model statistics are shown in *Figure 4—source data 2*. Reconstructed cryo-EM maps for T33_dn10, O43_dn18, and I53_dn5 and refined models are superimposed in *Figure 4*. Overall, the refined structures show excellent agreement with the corresponding Rosetta design models. Backbone r.m.s.d. values estimated for the asymmetric unit (consisting of a single subunit of component A and component B) were 0.65, 0.98, and 1.3 Å for T33_dn10, O43_dn18, and I53_dn5, respectively (*Table 1*).

**Figure 4.** Cryo-EM analysis of antigen-tailored nanoparticles. From left to right: cryo-EM maps with refined nanoparticle design models fit into electron density, view of designed nanoparticle interface region fit into cryo-EM density with indicated resolution (res.), designed nanoparticle 2D class-averages, raw cryo-EM micrographs of designed nanoparticles. (**a**) T33_dn10. (**b**) O43_dn18. (**c**) I53_dn5.

The online version of this article includes the following source data and figure supplement(s) for figure 4:

**Source data 1.** Cryo-EM data acquisition metrics for designed nanoparticles T33_dn10, O43_dn18, and I53_dn5.

**Source data 2.** Cryo-EM model building and refinement statistics for designed nanoparticles T33_dn10, O43_dn18, and I53_dn5.

**Figure supplement 1.** Cryo-EM data processing workflow.

### Characterization of viral glycoprotein-displaying nanoparticles

To explore the capability of the designed nanoparticles to present viral glycoproteins, we produced their trimeric fusion components genetically linked to a stabilized version of the BG505 SOSIP trimer. Synthetic genes for BG505 SOSIP fused to the N termini of T33_dn2A, T33_dn10A, and I53_dn5B (BG505 SOSIP–T33_dn2A, BG505 SOSIP–T33_dn10A, and BG505 SOSIP–I53_dn5B) were transfected into HEK293F cells. The secreted fusion proteins were then purified using a combination of immuno-affinity chromatography and SEC. The corresponding assembly component for each nanoparticle was produced recombinantly in *E. coli*, and *in vitro* assembly reactions were performed as equimolar mixtures of the two components overnight.

Assembled nanoparticles were purified by SEC and analyzed by NS-EM to assess particle assembly and homogeneity. ~ 1000 particles were manually picked and used to perform 2D classification and 3D classification/refinement in Relion (*Zivanov et al., 2018*). Models for the BG505 SOSIP-displaying nanoparticles fit into their reconstructed 3D maps are displayed in *Figure 5* (left). BG505 SOSIP trimers are clearly discernible in 2D class-averages and reconstructed 3D maps. However, the trimers appear less well-resolved than the corresponding nanoparticle core in the three reconstructions, likely due to the short flexible linkers between the BG505 SOSIP trimer and the fusion component. The self-assembling cores of the antigen-fused T33_dn2, T33_dn10, and I53_dn5 nanoparticles

**Figure 5.** NS-EM analysis of BG505 SOSIP-displaying nanoparticles. From left to right: BG505 SOSIP-displaying nanoparticle models fit into NS-EM density, 2D class-averages, raw NS-EM micrographs of assembled BG505 SOSIP-displaying nanoparticles. (a) BG505 SOSIP–T33_dn2. (b) BG505 SOSIP–T33_dn10. (c) BG505 SOSIP–I53_dn5.

The online version of this article includes the following figure supplement(s) for figure 5:

**Figure supplement 1.** Structural and antigenic characterization of DS-Cav1–I53_dn5.

**Figure supplement 2.** Structural and antigenic characterization of HA–I53_dn5.

were very similar to the NS-EM maps of the unmodified nanoparticles (at least to the resolution limits of NS-EM), demonstrating that fusion of the BG505 SOSIP trimer did not induce any major structural changes to the underlying nanoparticle scaffolds. Free components were detected in raw EM micrographs of BG505 SOSIP–I53_dn5, indicating some degree of disassembly. This finding is supported by stability data reported in a parallel study, where BG505 SOSIP–I53_dn5 demonstrated sensitivity to various physical and chemical stressors (*Antanasijevic et al., 2020*).

To further characterize the capability of the designed nanoparticles to present viral glycoproteins, we characterized the structures and antigenic profiles of I53_dn5 fused to the prefusion influenza HA and RSV F glycoproteins (HA–I53_dn5 and DS-Cav1–I53_dn5). Constructs were generated with each glycoprotein genetically linked to the N terminus of the I53_dn5B trimeric fusion component, and the proteins were secreted from HEK293F cells and purified by Ni$^{2+}$ IMAC. The fusion proteins were mixed with equimolar pentameric I53_dn5A for HA–I53_dn5 or I53_dn5A.1 (a stabilized and redox-insensitive variant of I53_dn5A lacking cysteines, see Materials and Methods) for DS-Cav1–I53_dn5, and the assembly reactions purified by SEC. For both assemblies, the majority of the material migrated in the peak expected for assembled nanoparticles, and NS-EM analysis showed formation of I53_dn5 nanoparticles with spikes emanating from the surface (*Figure 5—figure supplement 1* and *2*). In both cases, there was considerable variation in the spike geometry, again suggesting some flexibility between the glycoproteins and the underlying scaffold. The GG linker connecting DS-Cav1 to I53_dn5 likely accounts for the observed flexibility and suboptimal definition of the glycoprotein trimer in two-dimensional class averages (*Figure 5—figure supplement 1*, bottom right). There was no engineered linker between the glycoprotein and fusion component in the case of HA–I53_dn5, and more clearly defined spike density was observed in the class averages (*Figure 5—figure supplement 2*, bottom right).

To determine if the presented glycoproteins were properly folded, we examined their reactivity with conformation-specific monoclonal antibodies (mAbs). The DS-Cav1–I53_dn5 nanoparticle was found by an enzyme-linked immunosorbent assay (ELISA) to bind the RSV F-specific mAbs D25 (*McLellan et al., 2013b*), Motavizumab (*Cingoz, 2009*), and AM14 (*Gilman et al., 2015*) similarly to soluble DS-Cav1 trimer with foldon (*McLellan et al., 2013a*), indicating that the F protein is presented in the desired prefusion conformation on the nanoparticle (*Figure 4—figure supplement 1*, top). Biolayer interferometry binding experiments with anti-HA head - and stem-specific mAbs (*Krause et al., 2011*; *Ekiert et al., 2009*) analogously showed that both the HA–I53_dn5 nanoparticle and the HA–I53_dn5B trimer presented the head and stem regions with wild-type-like antigenicity (*Figure 5—figure supplement 2*, top).

## Tuning BG505 SOSIP epitope accessibility through nanoparticle presentation geometry

Previous work involving icosahedral nanoparticle scaffolds presenting HIV-1 Env trimers has shown that antigen crowding can modulate the accessibility of specific epitopes and thereby influence the humoral immune response (*Brouwer et al., 2019*; *Brinkkemper and Sliepen, 2019*). The nanoparticle scaffolds developed in this work were specifically designed to exhibit varying geometries and valencies, providing a unique way to manipulate and understand epitope accessibility in the context of nanoparticle vaccines. We selected the BG505 SOSIP–T33_dn2 tetrahedral nanoparticle (assembled *in vitro* using BG505 SOSIP–T33_dn2A and T33_dn2B) to compare against a previously published SOSIP-displaying icosahedral nanoparticle (BG505 SOSIP–I53-50) (*Brouwer et al., 2019*) through surface plasmon resonance (SPR) experiments. BG505 SOSIP–T33_dn2 presents four copies of the BG505 SOSIP trimer with much greater spacing than BG505 SOSIP–I53-50 with twenty copies. This difference derives from the angles between neighboring three-fold rotational symmetry axes— where the displayed BG505 trimers are located on the nanoparticle surfaces—in icosahedral and tetrahedral point group symmetries (41.8° and 109.5°, respectively). To first validate mAb binding to BG505 SOSIP–T33_dn2A, NS-EM class averages and a 3D reconstruction were generated in complex with the VRC01 Fab (*Walker et al., 2011*), confirming the expected binding mode (*Figure 6a*). Next, in part to simulate surface B cell receptors, a panel of anti-Env mAbs targeting epitopes ranging from the apex to the base of the BG505 SOSIP trimer were immobilized on SPR sensor chips (*Figure 6b*). BG505 SOSIP–T33_dn2A trimer or BG505 SOSIP–T33-dn2 nanoparticle was flowed over the mAbs and the ratio of macromolecule bound was calculated from the binding signal as previously described (*Brouwer et al., 2019*). For mAbs that target apical, V3-base, and CD4-binding site



**Figure 6.** BG505 SOSIP epitope accessibility compared between tetrahedral and icosahedral presentation geometries. (a) NS-EM micrographs of BG505 SOSIP–T33_dn2A with and without VRC01 Fab bound, 2D class averages, and models fit into NS-EM density. (b) Representative sensorgrams of indicated proteins binding to anti-Env mAbs. (c) Relative accessibility of epitopes on BG505 SOSIP–T33_dn2 nanoparticles and BG505 SOSIP–I53-50 nanoparticles as determined by mAb binding (top). Ratio of moles of macromolecules are means of 2–4 experimental replicates. Epitopes mapped

*Figure 6 continued on next page*

*Figure 6 continued*

onto BG505 SOSIP are presented on models of T33_dn2 and I53-50 (bottom). Wheat, antigen-fused trimeric component; purple, assembly component; gray, neighboring BG505 SOSIP trimers on the nanoparticle surface.

The online version of this article includes the following source data for figure 6:

**Source data 1.** BG505 SOSIP-T33_dn2 SPR Data.

epitopes (PGT145, PGT122, 2G12, and VRC01) (*Gilman et al., 2015*; *Walker et al., 2011*; *Trkola et al., 1996*; *Wu et al., 2010*), the number of molecules of trimer or nanoparticle bound was relatively similar (ratio ~ 1). However, for mAbs that target more base-proximate epitopes in the gp120-gp41 interface (ACS202, VRC34, and PGT151) (*van Gils et al., 2017*; *Kong et al., 2016*; *Falkowska et al., 2014*), an inter-protomeric gp41 epitope (3BC315) (*Klein et al., 2012*), and the main autologous neutralizing antibody epitope in the glycan hole centered on residues 241 and 289 (11B) (*McCoy et al., 2016*), the accessibility was reduced in the nanoparticle format. Binding to a mAb directed to the trimer base (12N) (*McCoy et al., 2016*) was not observed for nanoparticle BG505 SOSIP–T33_dn2 (*Figure 6b*). We compared epitope accessibility of BG505 SOSIP–T33_dn2 to that of BG505 SOSIP–I53-50 for six different mAbs (*Brouwer et al., 2019*). As for BG505 SOSIP–T33_dn2, mAbs to the apex, V3-base, and CD4-binding site (PGT145, PGT122, and VRC01) gave molar ratios of ~ 1 compared to BG505 SOSIP–I53-50. However, for mAbs that target the more base-proximate epitopes in the gp120-gp41 interface (VRC34 and PGT151), there was nearly three-fold higher epitope accessibility on T33_dn2 compared to I53-50 (*Figure 6c*). Further down the trimer, no accessibility difference was again observed for a mAb that targets the gp41 inter-protomeric epitope (3BC315), which was relatively inaccessible on both nanoparticles, likely due to steric hindrance by neighboring trimers. These findings demonstrate that antigen epitope accessibility can be finely tuned through presentation geometry, which could be used as a strategy to target the immune response against specific epitopes of interest.

## Discussion

Strong BCR signaling is required for eliciting robust humoral immune responses, but the molecular mechanisms by which this can be accomplished are not fully understood. Historically, live-attenuated or inactivated viruses and engineered virus-like particles (VLPs) have been able to confer protective immunity without pathogenicity, but the empirical discovery and compositional complexity of such vaccines has hampered understanding of possible mechanisms for obtaining sufficient levels of protective antibodies. *De novo* designed protein nanoparticles provide a modular way to present antigens to the immune system in defined geometries and of known composition. Multivalent antigen presentation can enhance antigen-specific antibody titers by orders of magnitude (*Bennett et al., 2015*; *Snapper, 2018*), but presentation of complex antigens is challenging due to the required geometric compatibility between antigen and scaffold. The design approach described here, in which nanoparticles incorporate *de novo* designed homo-oligomers tailored to present antigens of interest, is a general solution to this problem. More broadly, the ability to build protein-based nanomaterials with geometric specifications from scratch represents an important step forward in computational protein design, and provides a systematic way to investigate the influence of antigen presentation geometry on immune response.

The ability to fully tailor structures of nanoparticle scaffolds could be particularly useful for HIV-1 Env-based immunogens. While previous studies of HIV-1 Env trimers presented on nanoparticle scaffolds have demonstrated enhanced immunogenicity (*Escolano et al., 2019*), the effects are often modest compared to those observed for other antigens (*Bennett et al., 2015*; *Snapper, 2018*; *Marcandalli et al., 2019*; *Brinkkemper and Sliepen, 2019*). While there may exist intrinsic peculiarities to HIV-1 Env that limit increases in antibody responses upon multivalent presentation (*Klasse et al., 2012*; *Ringe et al., 2019*), limitations associated with epitope inaccessibility caused by crowding of the trimers on nanoparticle surfaces have also been identified (*Sanders and Moore, 2017*; *Brouwer et al., 2019*). This observation strongly motivates the exploration of antigen presentation across a range of scaffolds to identify geometries that most effectively elicit the desired immune response, particularly when it is of interest to direct the humoral immune response to specific epitopes. Indeed, the SPR experiments presented here demonstrate that epitopes proximate to

the BG505 SOSIP base were markedly more accessible to immobilized mAbs on BG505 SOSIP–T33_dn2 than BG505 SOSIP–I53-50, directly implicating steric crowding on the nanoparticle surface as a determinant of antigenicity. Furthermore, the availability of multiple antigen-displaying nanoparticles makes possible the usage of different scaffolds during prime and boost immunizations, which could limit immune responses directed toward the scaffolds while boosting antigen-specific antibody responses. Finally, the ability to tune antigen presentation geometry through *de novo* nanoparticle design provides a route to investigate the effects of this parameter on B cell activation, as well as the potency and breadth of the ensuing humoral response. This design approach could help overcome the intrinsically low affinity of germline BCRs for viral glycoproteins, and enable development of broadly neutralizing antibodies.

# Materials and methods

## Key resources table

| Reagent type (species) or resource | Designation | Source or reference | Identifiers | Additional information |
|---|---|---|---|---|
| Software, algorithm | RPX Method | PMID:28338692 | | Symmetric docking and scoring protocol |
| Software, algorithm | Sic_axle | PMID:30849373 | | Protein structure alignment protocol |
| Software, algorithm | Rosetta Macromolecular Modeling Suite | PMID:28430426 | RRID:SCR_015701 | Version 3 |
| Software, algorithm | Relion | PMID:23000701 | RRID:SCR_016274 | Cryo-EM structure determination software |
| Strain, strain background (*E. coli*) | BL21 | New England Biolabs | Cat. #:C2527H | Competent T7 expression strain |
| Strain, strain background (*E. coli*) | Lemo21 | New England Biolabs | Cat. #:C2527H | Competent T7 expression strain |
| Strain, strain background (*E. coli*) | HEK293F | PMID:26779721 | RRID:CVCL_6642 | Suspension-based cells for high yield expression of recombinant proteins |
| Chemical compound, drug | IPTG | Sigma | Cat. #:I6758 | Induces protein expression through T7 promoter |
| Chemical compound, drug | Kanamycin | Sigma | Cat. #:K1377 | Antibiotic |
| Chemical compound, drug | Carbenicillin | Sigma | Cat. #:C1389 | Antibiotic |
| Chemical compound, drug | Expifectamine | ThermoFisher | Cat. #:A38915 | Transfection reagent |
| Chemical compound, drug | Polyethylenimine | Polysciences Inc | Cat. #:23966 | Transfection reagent |
| Recombinant DNA reagent | pET21b(+) | Genscript | Addgene Cat. #:69741–3 | Bacterial expression vector |
| Recombinant DNA reagent | pET28b(+) | Gen9 | Addgene Cat. #:69865–3 | Bacterial expression vector |
| Recombinant DNA reagent | pPPI4 | Progenics Pharmaceuticals Inc PMID:10623724 | | Mammalian secretion vector, containing codon-optimized stabilized gp140 |

*Continued on next page*

*Continued*

| Reagent type (species) or resource | Designation | Source or reference | Identifiers | Additional information |
|---|---|---|---|---|
| Recombinant DNA reagent | CMV/R | PMID:15994776 | | Mammalian secretion vector, containing CMV enhancer/promoter with HTLV-1 R region |
| Antibody | PGT145 human monoclonal | PMID:21849977 | RRID:AB_2491054 | anti-HIV-1 Env (anti-Fc immobilization level of 320 ± 1.5 RU) |
| Antibody | PGT122 human monoclonal | PMID:21849977 | RRID:AB_2491042 | anti-HIV-1 Env (anti-Fc immobilization level of 320 ± 1.5 RU) |
| Antibody | 2G12 human monoclonal | PMID:8551569 | RRID:AB_2819235 | anti-HIV-1 Env (anti-Fc immobilization level of 320 ± 1.5 RU) |
| Antibody | VRC01 human monoclonal | PMID:20616233 | RRID:AB_2491019 | anti-HIV-1 Env (anti-Fc immobilization level of 320 ± 1.5 RU) |
| Antibody | ACS202 human monoclonal | PMID:27841852 | | anti-HIV-1 Env (anti-Fc immobilization level of 320 ± 1.5 RU) |
| Antibody | VRC34 human monoclonal | PMID:27174988 | RRID:AB_2819228 | anti-HIV-1 Env (anti-Fc immobilization level of 320 ± 1.5 RU) |
| Antibody | PGT151 human monoclonal | PMID:24768347 | | anti-HIV-1 Env (anti-Fc immobilization level of 320 ± 1.5 RU) |
| Antibody | 3BC315 human monoclonal | PMID:22826297 | | anti-HIV-1 Env (anti-Fc immobilization level of 320 ± 1.5 RU) |
| Antibody | 11B rabbit monoclonal | PMID:27545891 | | anti-HIV-1 Env (anti-Fc immobilization level of 320 ± 1.5 RU) |
| Antibody | 12N rabbit monoclonal | PMID:27545891 | | anti-HIV-1 Env (anti-Fc immobilization level of 320 ± 1.5 RU) |
| Antibody | 5J8 human monoclonal | PMID:21849447 | | anti-HA (20 µg/mL) |
| Antibody | CR6261 human monoclonal | PMID:19079604 | | anti-HA (20 µg/mL) |
| Antibody | D25 human monoclonal | PMID:24179220 | | anti-RSV F (1 pg/mL - 10 µg/mL) |
| Antibody | Motavizumab mouse-human chimeric monoclonal | PMID:20065632 | | anti-RSV F (1 pg/mL - 10 µg/mL) |
| Antibody | AM14 human monoclonal | PMID:26161532 | | anti-RSV F (1 pg/mL - 10 µg/mL) |

## Monomeric repeat proteins

Listed below are the RCSB Protein Data Bank entries for monomeric repeat proteins used for trimer docking and design in this study. An additional set of monomeric repeat proteins is provided in which experimental SAXS data agreed with the computational model (*Fallas et al., 2017*).

| X-ray Structures (PDB ID) | | SAXS Validated Models |
|---|---|---|
| 1na0 | (1NA0) | tpr1 |
| 3ltj | (3LTJ) | HR00 |
| 2fo7 | (2FO7) | |
| HR04 | (5CWB) | |
| HR07 | (5CWD) | |
| HR10 | (5CWG) | |

## Computational docking and design of antigen-tailored trimers

The monomeric repeat proteins were used as input to C3-symmetric trimer docking and design against the three viral antigens of interest: HIV-1 BG505 SOSIP, influenza H1 HA, and RSV F protein (PDB IDs 5VJ6 res. 518–664, 5KAQ res. 11–501, 5TPN res. 27–509) (*Wang et al., 2017*;

*Mousa et al., 2017*; *Joyce et al., 2016*). A C3-symmetric docking search was first performed, and output was assessed by the previously described RPX scoring method which discerns docks with more potential favorable pair-wise interactions than others (*Fallas et al., 2017*). Up to the top-scoring 100 docks for each repeat protein monomer were aligned against each of the three antigens along the shared C3 axis of symmetry and sampled translationally along and rotationally about the axis in 1 Å and 1° increments, respectively. For each sample, the distance between the C-terminal residue of the antigen and N-terminal residue of the docked trimer was measured until a minimum, non-clashing distance was determined (*Figure 1—figure supplement 1*). Solutions for docks that were less than or equal to 15 Å for one or more of the three antigens were selected for full Rosetta symmetric interface design as described in previously published methods (*Fallas et al., 2017*). Individual design trajectories were filtered by the following criteria: difference between Rosetta energy of bound (oligomeric) and unbound (monomeric) states less than −30.0 Rosetta energy units, interface surface area greater than 700 Å$^2$, Rosetta shape complementarity (sc) greater than 0.65, and less than 50 mutations made from the respective native monomer. Designs that passed these criteria were manually inspected and refined by single point reversions, and one design per unique docked configuration was added to the set of trimers selected for experimental validation.

## Computational docking and design of nanoparticles incorporating designed trimers

Two-component nanoparticle docks were scored and ranked using the RPX method (*Fallas et al., 2017*) as opposed to prior methods involving only interface residue contact count (*King et al., 2014*; *Bale et al., 2016*). High-scoring and non-redundant nanoparticle configurations were selected for Rosetta interface design with an added caveat that they include trimers with outward-facing N termini for antigen fusion. The design protocol took a single-chain input. pdb of each component, and a symmetry definition file (*DiMaio et al., 2011*) containing information for a specified cubic point group symmetry. The oligomers were then aligned to the corresponding axes of the symmetry using the Rosetta SymDofMover, taking into account the rigid body translations and rotations retrieved from the. dok file output from TCdock (*King et al., 2014*; *Bale et al., 2016*). A symmetric interface design protocol was employed which included pair-wise interaction motifs found from the RPX method (*Fallas et al., 2017*) within each Rosetta symmetric interface design trajectory (*King et al., 2014*; *Bale et al., 2016*). Individual design trajectories were filtered by the following criteria: difference between Rosetta energy of bound and unbound states less than −30.0 Rosetta energy units, interface surface area greater than 700 Å$^2$, sc greater than 0.6, and less than 50 mutations made from each native oligomer. Designs that passed these criteria were manually inspected and refined by single point reversions for mutations that did not appear to contribute to stabilizing the bound state of the interface. The sequence with the best overall metrics for each unique docked configuration was selected for experimental characterization.

## Designed trimer and nanoparticle protein expression and purification

Synthetic genes for designed proteins were optimized for *E. coli* expression and assembled from genes (purchased through Genscript or Gen9) ligated into the pET21b(+) (designed trimers) or pET28b(+) (designed nanoparticles) vector at restriction sites NdeI and XhoI or NcoI and XhoI, respectively. A second ribosome-binding site was inserted between the open-reading frames of individual components of nanoparticle designs ('AGAAGGAGATATCAT'), such that the two proteins would be co-expressed and screened for co-elution by SDS-PAGE. Plasmids were cloned into BL21 or Lemo21 (DE3) *E. coli* competent cells (New England Biolabs). Transformants were inoculated and grown in 5 mL of either LB or TB medium with either 100 mg/L carbenicillin or 100 µg/L kanamycin at 37°C overnight. Subsequently, liquid cultures were inoculated 1:100 (v:v) and grown at 37°C until an OD$_{600}$ of 0.5–0.8. Isopropyl-thio-β-D-galactopyranoside (IPTG) was then added at a concentration of 0.5–1 mM and growth temperature was reduced to 18°C to induce protein expression, or cultures were left at 37°C and auto-induced by media-included galactose according to the Studier protocols (*Studier, 2005*). Expression proceeded for 20 hr until the cell cultures were harvested by centrifugation. Cell pellets were resuspended in 25 mM Tris, 150 mM NaCl, 5 mM imidazole, DNase, EDTA-free protease inhibitors (Pierce), pH 8.0. and lysed by sonication or microfluidization. Each protein was then purified from lysate by Ni$^{2+}$ IMAC with Ni-NTA Superflow resin (Qiagen or GE). Resin with

bound cell lysate was washed with 15 column volumes of 25 mM Tris, 150 mM NaCl, 40 mM imidazole, pH 8.0. Proteins were eluted with five column volumes of 25 mM Tris, 150 mM NaCl, 400 mM imidazole, pH 8.0 for further purification by SEC.

### Designed trimer and nanoparticle SEC

Elution samples for each designed protein were concentrated down using a 10,000 MWCO protein concentrator (Novagen) and fractionated by size on an AKTA pure chromatography system using a Superdex 200 (for designed trimers) or Superose 6 10/300 GL column (for designed nanoparticles) in 25 mM Tris, 150 mM NaCl, pH 8.0 (TBS). Sizing profiles were noted based on absorption at 220 nm and 280 nm wavelength light for each fraction. Molecular weights for predominant species in each protein trace were estimated by comparison to the corresponding monomeric profile.

### Designed trimer and nanoparticle SEC-MALS

Fractions containing single predominant species from an initial SEC purification were concentrated down with 10,000 MWCO protein concentrators (Novagen) to a concentration of 1.0–2.0 mg/mL and run through a high-performance liquid chromatography system (Agilent) using a Superdex 200 or Superose 6 10/300 GL column (GE Life Sciences) in TBS buffer. These fractionation runs were coupled to a multi-angle light scattering detector (Wyatt) to determine the absolute molecular weights for each designed protein complex.

### Designed trimer and nanoparticle SAXS

Designed proteins that predominantly formed the target oligomeric species were re-expressed and purified for low-resolution solution structure determination by SAXS at the SIBYLS High Throughput SAXS Advanced Light Source in Berkeley, California (*Dyer et al., 2014*). A beam exposure time of between 0.3 and 10 s was used to obtain averaged diffraction data (SAXS FrameSlice Application), which are represented in plots of log intensity (I) vs. q. A 11kEV/1.125A X-ray beam was used with a 2 m beamstop.

### Designed trimer crystallization conditions

Design 1na0C3_2 was found to crystallize in 1 M LiCl, 100 mM citrate, 20% w/v PEG 6000, pH 4, and was frozen using 25% glycerol as cryoprotectant. Design 3ltjC3_1 crystallized in 1 mM DL-glutamic acid monohydrate, 100 mM DL-alanine, 100 mM glycine, 100 mM DL-lysine monohydrochloride, 100 mM DL-serine; 100 mM Tris, 100 mM BICINE, 20% v/v ethylene glycol, 10% w/v PEG 8000, pH 8.5. Diffraction data for each of these designs were collected at the Advanced Light Source (Beamline 8.2.1) at Lawrence Berkeley National Laboratory in Berkeley, California. Both designed trimers contained uncleaved C-terminal $His_6$-tags in crystallized conditions.

### Designed trimer crystal diffraction data collection, structure determination, and refinement

Diffraction data for 3ltjC3_1 was collected on beamline 5.0.1 at the Advanced Light Source (Berkeley, CA) and 1na0C3_2 on beamline 8.2.1, both using an ADSC Q315R CCD area detector. Both datasets were scaled and merged in HKL2000 (*Otwinowski and Minor, 1997*). The structures were phased by molecular replacement, with the computational design as the search model, using the program PHASER (*McCoy et al., 2007*) in the PHENIX software suite (*Adams, 2012*). Iterative rounds of manual model building and refinement were conducted in Coot (*Emsley and Crispin, 2018*) and Phenix.refine (*Afonine et al., 2012*), respectively for both structures. Hydrogens were added for all refinement runs. The geometric quality of the final model was assessed using the Molprobity server (*Chen et al., 2010*). Resolution cutoff was determined by monitoring the refinement statistics in the context of the reflection data completeness and the $CC^{1/2}$ and $I/\sigma I$ values (*Karplus and Diederichs, 2012*).

### I53_dn5 *in vitro* assembly

Genes for the individual nanoparticle components were cloned into expression vectors and expressed independently in *E. coli*. The $His_6$-tagged proteins were purified following the purification protocol described for the designed trimers. Initial SEC chromatograms for the components were

obtained on a Superdex 200 10/300 GL column, and predominant peak species were stoichiometrically mixed in TBS buffer for 20 min at 25°C. A secondary SEC step was performed on a Superose 6 10/300 GL column to assess assembly of the intended particle based on expected retention volume.

## Design of cysteine-free I53_dn5A.1 pentamer

Multiple rounds of designs were performed to remove native unpaired cysteines from I53_dn5A. In the first round of design, cysteines were mutated to alanines (C94A, C119A), which caused the protein to bind and retain through purification a bright yellow metabolite. Further mutations were introduced to knock out metabolite binding in the native enzymatic active site (W18G), which led to protein precipitation during purification. Additional mutations were made (K84R, M88P, E91D, L117I, L120D) to re-stabilize the protein, named I53_dn5A.1.

## Production and purification of BG505 SOSIP–T33_dn2A, BG505 SOSIP–T33_dn10A, and BG505 SOSIP–I53_dn5B

Synthetic genes were optimized for mammalian expression and subcloned into the pPPI4 vector. BamHI and NheI restriction sites were used for insertion of different nanoparticle components to the C terminus of BG505 SOSIP. Quick Ligation kit, BamHI-HF, and NheI-HF restriction enzymes were purchased from New England Biolabs. BG505 SOSIP variant used for all early optimizations steps was engineered with a combination of v5.2 (*Torrents de la Peña et al., 2017*) (mutations: E64K, A73C, A316W, A561C) and MD3D (*Steichen et al., 2016*) (mutations: M271I, A318Y, R585H, L568D, V570H, R304V, F519S) stabilizing mutations, and had glycosylation sites introduced at positions 241 and 289 (mutations: P240T, S241N, F288L, T290E, P291S). This construct was termed BG505 SOSIP. v5.2 (7S). For epitope-accessibility experiments (by surface plasmon resonance), a version of this construct was designed without the 241 and 289 glycans. HEK 293F (RRID:CVCL_6642) cells were grown in suspension using FreeStyle 293 Expression Medium (Thermo Fisher Scientific) at 135 RPM, 8% $CO_2$, 80% humidity, 37°C. At confluency of ~ $1 \times 10^6$ cells/ml, the cells were co-transfected with pPPI4 DNA vectors encoding the appropriate fusion component (250 µg per 1 L of cells) and furin protease (80 µg per 1 L of cells). Polyethylenimine (Polysciences Inc) was used as a transfection reagent (1 mg per 1 L of cells). Cells were incubated for 6 days, after which they were spun down by centrifugation (7,000 RPM, 1 hr, 4°C) and the protein-containing supernatant was further clarified by vacuum-filtration (0.45 µm, Millipore Sigma). For immuno-affinity chromatography steps, Sepharose 4B columns with immobilized PGT145 IgG were used (RRID:AB_2491054). Fusion components were eluted with 3 M magnesium chloride, 250 mM L-Arginine buffer, pH 7.2 into an equal volume of SEC buffer (25 mM Tris, 250 mM L-Arginine, 500 mM NaCl, 5% glycerol, pH 7.4). Eluates were concentrated and buffer exchanged into SEC buffer. A Sephacryl S200 16/600 column was used for subsequent SEC purification.

## Production and Purification of HA–I53_dn5B

Synthetic genes were optimized for mammalian expression and subcloned into the CMV/R vector (VRC 8400) (*Barouch et al., 2005*). XbaI and AvrII restriction sites were used for insertion of I53_dn5B component to the C terminus of the H1 HA ectodomain (residues 1–676 from A/Michigan/45/2015), which also contained a Y98F mutation to prevent sialic-acid binding and self-aggregation during expression (*Whittle et al., 2014*). Gene synthesis and cloning was performed by Genscript. HEK 293 F cells were grown in suspension using Expi293 Expression Medium (Thermo Fisher Scientific) at 150 RPM, 8% $CO_2$, 70% humidity, 37°C. At confluency of ~ $2.5 \times 10^6$ cells/mL, the cells were co-transfected with the vector encoding HA–I53_dn5B (1000 µg per 1 L of cells). Expifectamine was used as a transfection reagent according to the manufacturer's protocol. Cells were incubated for 96 hr, after which they were spun down by centrifugation (4,000 RPM, 20 min, 4°C) and the protein-containing supernatant was further clarified by vacuum-filtration (0.45 µm, Millipore Sigma). For nickel-affinity chromatography steps, a background of 50 mM Tris, 350 mM NaCl, pH 8.0 was added to clarified supernatant. For each liter of supernatant, 4 mL of Ni Sepharose excel resin (GE) was rinsed into phosphate-buffered saline (PBS) using a gravity column and then added to the supernatant, followed by overnight shaking at 4°C. After 16–24 hr, resin was collected and separated from the mixture and washed twice with 50 mM Tris, 500 mM NaCl, 30 mM imidazole, pH 8.0 prior to elution of desired protein with 50 mM Tris, 500 mM NaCl, 300 mM imidazole, pH 8.0. Eluates were

concentrated and applied to a HiLoad 16/600 Superdex 200 pg column pre-equilibrated with PBS for purification by SEC.

## Production and Purification of DS-Cav1–I53_dn5

Gene synthesis and cloning was performed by Genscript. HEK 293 F cells (RRID:CVCL_6642) were grown in suspension using Expi293 expression medium (Thermo Fisher Scientific) at 150 RPM, 8% $CO_2$, 70% humidity, 37°C. At confluency of ~ 2.5 to $3 \times 10^6$ cells/ml, the cells were transiently transfected with the vector encoding DS-Cav1–I53_dn5B (1 mg per 1 L of cells). Expifectamine was used as a transfection reagent according to the manufacturer's protocol. Cells were incubated for 96 hr and spun down by centrifugation (4,000 RPM for 20 min at 4°C). Supernatant was vacuum-filtered (0.45 μm, Millipore Sigma) and 50 mM Tris, 350 mM NaCl, pH 8.0 was added for nickel-affinity chromatography. Ni Sepharose resin (GE) was washed three times with PBS by centrifugation (2,000 RPM for 5 min at 4°C) and added to the supernatant. Nickel-supernatant was incubated either overnight at 4°C or for 2 hr at room temperature. Resin was collected and separated from the mixture and washed twice with 50 mM Tris, 500 mM NaCl, 30 mM imidazole, pH 8.0 prior to elution of desired protein with 50 mM Tris, 500 mM NaCl, 300 mM imidazole, pH 8.0. Eluates were concentrated and applied to a HiLoad 10/300 Superdex 200 Increase GL column pre-equilibrated with PBS for purification by SEC.

## Assembly and purification of antigen-displaying nanoparticles

Several reactions containing 5–10 μg of the purified fusion component and an equimolar amount of the corresponding assembly component were prepared and incubated under different conditions (varying temperature and assembly buffer) for 24 hr. Native PAGE Bis-Tris gels (Thermo Fisher Scientific) and NS-EM was used for detection of assembly. Following the identification of optimal assembly conditions, milligram quantities of particles were assembled and purified by SEC (Superose six or Sepharose 500 column) with TBS as the running buffer. Fractions corresponding to the fusion component were pooled and concentrated (Amicon Ultra Centrifugal Filter Units, Millipore Sigma).

## NS-EM of T33_dn2, T33_dn5, T33_dn10, O43_dn18, I53_dn5, BG505 SOSIP–T33_dn2, BG505 SOSIP–T33_dn10, and BG505 SOSIP–I53_dn5

NS-EM experiments were performed as described previously (*Lee and Gui, 2016*; *Ozorowski et al., 2018*). Fusion components and assembled nanoparticle samples (with and without antigen) were diluted to 20–50 μg/ml and loaded onto the carbon-coated 400-mesh Cu grid that had previously been glow- discharged at 15 mA for 25 s. VRC01 (RRID:AB_2491019) complexes with BG505 SOSIP–T33_dn2A were formed by combining the trimer with a six-fold molar excess of the VRC01 Fab and subsequent incubation for 1 hr at room temperature. Complex sample was diluted to 20 μg/ml and loaded onto the glow discharged Cu grids. Grids were negatively stained with 2% (w/v) uranyl-formate for 60 s. Data collection was performed on a Tecnai Spirit electron microscope operating at 120 keV. The magnification was 52,000 × with a pixel size of 2.05 Å at the specimen plane. The electron dose was set to 25 e⁻/Å (*Snapper, 2018*). All imaging was performed with a defocus value of −1.50 μm. The micrographs were recorded on a Tietz 4k × 4 k TemCam-F416 CMOS camera using Leginon automated imaging interface. Data processing was performed in Appion data processing suite. For BG505 SOSIP-fused nanoparticle samples (v5.2 (7S)), approximately 500–1000 particles were manually picked from the micrographs and 2D-classified using the Iterative MSA/MRA algorithm. For non-fused nanoparticle samples, 2,000–5000 particles were manually picked and processed. For BG505 SOSIP-fused trimer samples and Fab complexes, 10,000–40,000 particles were auto-picked and 2D-classified using the iterative MSA/MRA algorithm. 3D classification and refinement steps were done in Relion/2.1 (RRID:SCR_015701) (*Kimanius et al., 2016*). The resulting EM maps have been deposited to EMDB with IDs: 21162 (T33_dn2), 21163 (T33_dn5), 21164 (T33_dn10), 21165 (O43_dn18), 21166 (I53_dn5), 21167 (BG505 SOSIP–T33_dn2A), 21168 (BG505 SOSIP–T33_dn2A + VRC01 Fab), 21169 (BG505 SOSIP–T33_dn2 nanoparticle), 21170 (BG505 SOSIP–T33_dn10 nanoparticle), 21171 (BG505 SOSIP–I53_dn5 nanoparticle).

### NS-EM of HA–I53_dn5

The HA–I53_dn5 complex was adsorbed onto glow-discharged carbon-coated copper mesh grids for 60 s, stained with 2% uranyl formate for 30 s, and allowed to air dry. Grids were imaged using the FEI Tecnai Spirit 120 kV electron microscope equipped with a Gatan Ultrascan 4000 CCD Camera. The pixel size at the specimen level was 1.60 Å. Data collection was performed using Leginon (*Suloway et al., 2005*) with the majority of the data processing carried out in Appion (*Lander et al., 2009*). The parameters of the contrast transfer function (CTF) were estimated using CTFFIND4 (*Mindell and Grigorieff, 2003*). All particles were picked in a reference-free manner using DoG Picker (*Voss et al., 2009*). The HA–I53_dn5 particle stack from the micrographs collected was pre-processed in Relion (RRID:SCR_015701). Reference-free two-dimensional (2D) classification with cryoSPARC was used to select a subset of particles, which were used to generate an initial model using the Ab-Initio reconstruction function in CryoSPARC. The particles from the best class were used for non-uniform refinement in CryoSPARC to obtain the final 3D reconstruction.

### NS-EM of DS-Cav1–I53_dn5

The sample was diluted with a buffer containing 10 mM HEPES pH 7.0 and 150 mM NaCl to a concentration of 0.025 mg/ml and adsorbed for 15 s to a glow-discharged carbon-coated copper grid. The grid was washed with the same buffer and stained with 0.7% uranyl formate. Images were collected at a nominal magnification of 57,000 × using EPU software on a ThermoFisher Talos F200C electron microscope equipped with a 4k × 4 k Ceta camera and operated at 200 kV. The pixel size was 0.253 nm. Particles were picked automatically using in-house written software (unpublished) and extracted into 200 × 200 pixel boxes. Reference-free 2D classifications were performed using Relion 1.4 (*Zivanov et al., 2018*) and SPIDER (*Frank et al., 1996*).

### Cryo-EM of designed nanoparticles T33_dn10, O43_dn18, and I53_dn5

Grids were prepared on Vitrobot mark IV (Thermo Fisher Scientific). Temperature was set to 10°C, humidity at 100%, wait time at 10 s, while the blotting time was varied in the 4–7 s range with the blotting force at 0. The concentrations of T33_dn10, O43_dn18, and I53_dn5 nanoparticle samples were 4.2, 3.0, and 1.9 mg/ml, respectively. n-Dodecyl β-D-maltoside (DDM) at a final concentration of 0.06 mM was used for sample freezing. Quantifoil R 2/1 holey carbon copper grids (Cu 400 mesh) were pre-treated with $Ar/O_2$ plasma (Solarus plasma cleaner, Gatan) for 10 s prior to sample application. Concentrated nanoparticle samples were mixed with appropriate volumes of stock DDM solution and 3 µl applied onto the grid. Excess sample and buffer was blotted off and the grids were plunge-frozen into nitrogen-cooled liquid ethane. Cryo-grids were loaded into a Titan Krios (FEI) operating at 300 kV, equipped with the K2 direct electron detector camera (Gatan). Exposure magnification of 29,000 was set with the resulting pixel size of 1.03 Å at the specimen plane. Total dose was set to ∼ 50 e⁻/Å (*Snapper, 2018*) with 250 ms frames. Nominal defocus range was set to −0.6 to −1.6 µm for all three nanoparticle samples. Automated data collection was performed using Leginon software (*Suloway et al., 2005*). Data collection information for acquired datasets is shown in *Figure 4—source data 1*.

### Cryo-EM data processing

MotionCor2 (*Zheng et al., 2017*) was run to align and dose-weight the movie micrographs. GCTF v1.06 was applied to estimate the CTF parameters. Initial processing was performed in cryoSPARC 2.9.0. Template-picked particles were extracted and subjected to 2D classification. Multiple rounds of heterogeneous refinement were performed to further clean-up particle stacks in three acquired datasets. Selected subsets of particles were then transferred to Relion 3.0 (RRID:SCR_015701) (*Zivanov et al., 2018*) for further processing. Reference models were generated using Ab-Initio reconstruction in cryoSPARC v2.9.0 (*Punjani et al., 2017*) with the application of appropriate symmetry (tetrahedral, octahedral, and icosahedral for T33_dn10, O43_dn18, and I53_dn5, respectively). Several rounds of 3D classification and refinement were used to sort out a subpopulation of particles that went into the final 3D reconstructions. Tetrahedral, octahedral, and icosahedral symmetry restraints were applied for all 3D refinement and classification steps during the processing of T33_dn10, O43_dn18, and I53_dn5 datasets, respectively. Soft solvent mask around the nanoparticle core was introduced during the final 3D classification, refinement, and post-processing. The

resolutions of the final reconstructed maps were 3.86 Å for T33_dn10, 4.54 Å for O43_dn18, and 5.35 Å for I53_dn5. The resulting EM maps have been deposited to EMDB with IDs: 21172 (T33_dn10), 21173 (O43_dn18) and 21174 (I53_dn5). A graphical summary of the data processing approach and relevant information for each dataset are displayed in *Figure 4—figure supplement 1*.

## SPR Analysis of BG505 SOSIP-fused Trimer and Nanoparticle Binding to Immobilized mAbs

The antigenicity of BG505 SOSIP–T33_dn2A trimer and BG505 SOSIP–T33_dn2 nanoparticle was analyzed on a BIAcore 3000 instrument at 25 °C and with HBS-EP (GE healthcare Life sciences) as running buffer, as described (*Brouwer et al., 2019*). Affinity-purified goat anti-human IgG Fc (Bethyl Laboratories, Inc) and goat anti-rabbit IgG Fc (Abcam) were amine-coupled to CM3 chips and the anti-HIV-1 Env human and rabbit mAbs were captured to an average density of 320 ± 1.5 RU (s.e.m.). BG505 SOSIP–T33_dn2A or BG505 SOSIP–T33_dn2 (both v5.2(7S) without N241/N289) (*Torrents de la Peña et al., 2017*; *Steichen et al., 2016*) was allowed to associate for 300 s and then dissociate for 600 s at a concentration of 10 nM assembled macromolecule (trimer or nanoparticle). The low background binding in parallel flow cells with only anti-Fc was subtracted. The lack of binding of nanoparticles lacking Env was ascertained for each mAb. To illustrate how epitope accessibility affects the relative binding of the trimers and nanoparticles, we converted the signals, which are proportional to mass bound, to moles bound and calculated the ratio for nanoparticles/trimers. For this comparison historic data on icosahedral nanoparticles were included (*Brouwer et al., 2019*). The number of moles binding to the immobilized IgG at the end of the association phase was calculated: $n = \frac{R \cdot m \cdot A}{M}$ where $n$ is the number of moles of macromolecules, $R$ the response at 300 s (RU), $m$ the mass bound per area and RU (g/(mm [*Snapper, 2018*] RU)), $A$ the interactive area of the chip (mm [*Snapper, 2018*]), and $M$ the molar mass of the macromolecule (g/mol). This analysis corrects for the greater mass (and thereby greater signal) for each bound nanoparticle such that the number of binding events by differing macromolecules can be directly compared.

## Biolayer Interferometry on HA–I53_dn5

To produce biotin-labeled antibodies specific to the H1 HA head, mAb 5J8 (*Krause et al., 2011*) in PBS was mixed with a 20 × molar excess (relative to complete antibodies) of EZ-Link NHS-LC-Biotin (Thermo Fisher Scientific) and allowed to sit for 2 hr at 4°C, followed by two rounds of overnight dialysis against PBS at 4°C to remove excess biotinylation reagent. All biosensors were hydrated in assay buffer (25 mM Tris, 150 mM NaCl, 0.5% bovine serum albumin, 0.01% TWEEN 20, pH 8.0) before use. Biotinylated 5J8 (20 μg/mL in assay buffer) was immobilized on SA biosensors, then briefly dipped in assay buffer prior to exposure to designed H1 HA fusions (500 nM per asymmetric unit, in assay buffer). The biosensor was again dipped in assay buffer and then exposed to the stem-specific mAb CR6261 (20 μg/mL in assay buffer) (*Throsby et al., 2008*).

## Analytical SEC on HA–I53_dn5

Purified HA-displaying nanoparticles or trimers were applied to a Sephacryl S-500 column pre-equilibrated with 25 mM Tris, 2 M NaCl, 5% glycerol, pH 8.0. Sizing profiles were recorded based on absorption at 280 nm wavelength light.

## ELISA Assays on DS-Cav1–I53_dn5

ELISA was used to measure binding kinetics of DS-Cav1–I53_dn5 to RSV F-specific mAbs D25, Motavizumab, and AM14. D25 is a prefusion specific mAb that binds site Ø (*McLellan et al., 2013b*). Motavizumab binds site II of the pre and post-fusion conformations (*Cingoz, 2009*). AM14 is trimer-specific binding across protomers of the prefusion conformation (*Gilman et al., 2015*). 96-well ELISA plates were coated with 2 μg/mL DS-Cav1 nanoparticles. Plates were incubated at 4°C overnight and blocked with PBS containing 5% skim milk at 37°C for 30 min. mAbs listed were serially diluted in fourfold steps, and then added to the plates and incubated at 37°C for 45 min. Horseradish peroxidase (HRP)-conjugated anti-human IgG (Southern Biotech., Birmingham, AL) was added and incubated at 37°C for 30 min, followed by 3,3′,5′,5- Tetramethylbenzidine (TMB; Sigma-Aldrich, St. Louis,

MO) HRP substrate, and yellow color that developed after the addition of 1 M $H_2SO_4$ was measured by absorbance at 450 nm.

## Acknowledgements

This work was supported by the Howard Hughes Medical Institute, the Bill and Melinda Gates Foundation (OPP1120319, OPP1111923, OPP1156262, OPP1115782 and OPP1084519), the National Science Foundation (NSF CHE 1629214), The Audacious Project at the Institute for Protein Design, and the Intramural Research Program of the Vaccine Research Center, NIAID, NIH. We thank Lauren Carter at the Institute for Protein Design for assistance with SEC-MALS. We thank M Capel, K Rajashankar, N Sukumar, J Schuermann, I Kourinov and F Murphy at NECAT supported by grants from the National Center for Research Resources (5P41RR015301-10) and the National Institute of General Medical Sciences (P41 GM103403-10) from the National Institutes of Health. We thank Kathryn Burnett and Greg Hura for SAXS data collection through the SIBYLS mail-in SAXS program. This work was conducted at the Advanced Light Source (ALS), a national user facility operated by Lawrence Berkeley National Laboratory on behalf of the Department of Energy, Office of Basic Energy Sciences, through the Integrated Diffraction Analysis Technologies (IDAT) program, supported by DOE Office of Biological and Environmental Research. Additional support comes from the National Institute of Health project ALS-ENABLE (P30 GM124169) and a High-End Instrumentation Grant S10OD018483. X-ray crystallography data were collected at the ALS, and the Berkeley Center for Structural Biology is supported in part by the National Institutes of Health, National Institute of General Medical Sciences, and the Howard Hughes Medical Institute. The ALS is supported by the Director, Office of Science, Office of Basic Energy Sciences, of the U.S. Department of Energy under Contract No. DE-AC02-05CH11231.

## Additional information

### Competing interests

George Ueda, Jorge A Fallas: Inventor on U.S. patent application 62/422,872 titled "Computational design of self-assembling cyclic protein homo-oligomers." Inventor on U.S. patent application 62/636,757 titled "Method of multivalent antigen presentation on designed protein nanomaterials." Inventor on U.S. patent application PCT/US20/17216 titled "Nanoparticle-based Influenza Virus Vaccines and Uses Thereof.". William Sheffler: Inventor on U.S. patent application 62/422,872 titled "Computational design of self-assembling cyclic protein homo-oligomers.". Daniel Ellis: Inventor on U.S. patent application 62/636,757 titled "Method of multivalent antigen presentation on designed protein nanomaterials." Inventor on U.S. patent application PCT/US20/17216 titled "Nanoparticle-based Influenza Virus Vaccines and Uses Thereof.". Masaru Kanekiyo: Inventor on U.S. patent application PCT/US20/17216 titled "Nanoparticle-based Influenza Virus Vaccines and Uses Thereof.". Barney S Graham: Inventor on U.S. patent application PCT/US20/17216 titled "Nanoparticle-based Influenza Virus Vaccines and Uses Thereof." Member of Icosavax's Scientific Advisory Board. Neil P King: Inventor on U.S. patent application 62/636,757 titled "Method of multivalent antigen presentation on designed protein nanomaterials." Inventor on U.S. patent application PCT/US20/17216 titled "Nanoparticle-based Influenza Virus Vaccines and Uses Thereof." Co-founder and shareholder of Icosavax, a company that has licensed these patent applications. Member of Icosavax's Scientific Advisory Board. David Baker: Inventor on U.S. patent application 62/422,872 titled "Computational design of self-assembling cyclic protein homo-oligomers." Inventor on U.S. patent application 62/636,757 titled "Method of multivalent antigen presentation on designed protein nanomaterials." Inventor on U.S. patent application PCT/US20/17216 titled "Nanoparticle-based Influenza Virus Vaccines and Uses Thereof." Co-founder and shareholder of Icosavax, a company that has licensed these patent applications. Member of Icosavax's Scientific Advisory Board. The other authors declare that no competing interests exist.

## Funding

| Funder | Grant reference number | Author |
| --- | --- | --- |
| Bill and Melinda Gates Foundation | OPP1120319 | George Ueda<br>Jorge A Fallas<br>William Sheffler<br>Daniel Ellis<br>Adam Moyer<br>Matthew J Bick<br>Neil P King<br>David Baker |
| Bill and Melinda Gates Foundation | OPP1111923 | George Ueda<br>Jorge A Fallas<br>William Sheffler<br>Daniel Ellis<br>Adam Moyer<br>Matthew J Bick<br>Neil P King<br>David Baker |
| Bill and Melinda Gates Foundation | OPP1156262 | George Ueda<br>Jorge A Fallas<br>William Sheffler<br>Daniel Ellis<br>Adam Moyer<br>Matthew J Bick<br>Neil P King<br>David Baker |
| Bill and Melinda Gates Foundation | OPP1115782 | Andrew B Ward |
| Bill and Melinda Gates Foundation | OPP1084519 | Andrew B Ward |
| National Science Foundation | NSF CHE 1629214 | George Ueda<br>Jorge A Fallas<br>William Sheffler<br>Daniel Ellis<br>Adam Moyer<br>Matthew J Bick<br>Neil P King<br>David Baker |
| National Center for Research Resources | 5P41RR015301-10 | George Ueda<br>Jorge A Fallas<br>William Sheffler<br>Daniel Ellis<br>Adam Moyer<br>Matthew J Bick<br>Neil P King<br>David Baker |
| National Institute of General Medical Sciences | P41 GM103403-10 | George Ueda<br>Jorge A Fallas<br>William Sheffler<br>Daniel Ellis<br>Adam Moyer<br>Matthew J Bick<br>Neil P King<br>David Baker |
| National Institute of General Medical Sciences | P30 GM124169-01 | Banumathi Sankaran<br>Peter H Zwart |
| National Institute of General Medical Sciences | High-End Instrumentation Grant S10OD018483 | Banumathi Sankaran<br>Peter H Zwart |
| U.S. Department of Energy | Contract No. DE-AC02-05CH11231 | Banumathi Sankaran<br>Peter H Zwart |

| | | |
|---|---|---|
| The Audacious Project | | George Ueda<br>Jorge A Fallas<br>William Sheffler<br>Daniel Ellis<br>Adam Moyer<br>Matthew J Bick<br>Neil P King<br>David Baker |
| National Institute of Allergy and Infectious Diseases | HIVRAD P01 AI 110657 | John P Moore<br>Per Johan Klasse<br>Anila Yasmeen |

The funders had no role in study design, data collection and interpretation, or the decision to submit the work for publication.

## Author contributions

George Ueda, Conceptualization, Data curation, Software, Formal analysis, Funding acquisition, Validation, Investigation, Visualization, Methodology, Writing - original draft, Writing - review and editing, Wrote the paper, Designed the research, Performed computational docking, viral glycoprotein modeling, and design calculations, Experimentally produced and characterized designed trimers and nanoparticles through SEC-MALS, SAXS, and crystallography, Assisted with design of antigen-displaying nanoparticles, Prepared the manuscript; Aleksandar Antanasijevic, Conceptualization, Data curation, Software, Formal analysis, Validation, Investigation, Visualization, Methodology, Writing - review and editing, Assisted with writing the paper, Assisted with research design, Performed cryo-EM analysis on designed nanoparticles, Produced and characterized BG505 SOSIP-displaying nanoparticles, Assisted with preparing the manuscript; Jorge A Fallas, Conceptualization, Software, Formal analysis, Supervision, Validation, Investigation, Visualization, Methodology, Writing - review and editing, Implemented code for the RPX method into the Rosetta software suite, Assisted with preparing the manuscript; William Sheffler, Conceptualization, Software, Validation, Visualization, Methodology, Assisted with implementing code for the RPX method into the Rosetta software suite; Jeffrey Copps, Data curation, Software, Formal analysis, Validation, Visualization, Writing - review and editing, Performed NS-EM analysis on designed nanoparticles; Daniel Ellis, Data curation, Formal analysis, Validation, Investigation, Visualization, Writing - review and editing, Assisted with production and characterization of HA-I53_dn5, Assisted with preparing the manuscript; Geoffrey B Hutchinson, Data curation, Formal analysis, Validation, Investigation, Visualization, Writing - review and editing, Assisted with production and characterization of DS-Cav1-I53_dn5, Assisted with preparing the manuscript; Adam Moyer, Data curation, Validation, Investigation, Visualization, Assisted with nanoparticle design; Anila Yasmeen, Data curation, Formal analysis, Validation, Investigation, Visualization, Writing - review and editing, Performed SPR experiments and assisted with quantitative comparison between BG505 SOSIP-displaying nanoparticles; Yaroslav Tsybovsky, Data curation, Formal analysis, Validation, Investigation, Visualization, Writing - review and editing, Performed NS-EM analysis on DS-Cav1, Assisted with preparing the manuscript; Young-Jun Park, Data curation, Formal analysis, Validation, Investigation, Visualization, Writing - review and editing, Performed NS-EM analysis on HA-I53_dn5, Assisted with preparing the manuscript; Matthew J Bick, Data curation, Formal analysis, Validation, Investigation, Visualization, Writing - review and editing, Processed diffraction data and solved the crystal structure of 3ltjC3_1v2, Assisted with preparing the manuscript; Banumathi Sankaran, Data curation, Formal analysis, Validation, Investigation, Visualization, Writing - review and editing, Processed diffraction data and solved the crystal structure of 1na0C3_2, Assisted with preparing the manuscript; Rebecca A Gillespie, Data curation, Formal analysis, Validation, Investigation, Visualization, Assisted with production and characterization of DS-Cav1-I53_dn5; Philip JM Brouwer, Formal analysis, Investigation, Visualization, Writing - review and editing, Assisted with quantitative comparison bewteen BG505 SOSIP-displaying nanoparticles, Assisted with preparing the manuscript; Peter H Zwart, Formal analysis, Validation, Visualization, Assisted with solving the crystal structure of 1na0C3_2; David Veesler, Formal analysis, Supervision, Validation, Visualization, Assisted with NS-EM analysis on HA-I53_dn5; Masaru Kanekiyo, Conceptualization, Formal analysis, Supervision, Validation, Visualization, Methodology, Writing - review and editing, Assisted with production and characterization of DS-Cav1-I53_dn5, Assisted with preparing the manuscript; Barney S Graham, Conceptualization, Formal analysis, Supervision, Funding acquisition, Validation,

Visualization, Methodology, Assisted with production and characterization of DS-Cav1-I53_dn5; Rogier W Sanders, Conceptualization, Formal analysis, Supervision, Validation, Visualization, Methodology, Assisted with quantitative comparison bewteen BG505 SOSIP-displaying nanoparticles; John P Moore, Conceptualization, Formal analysis, Supervision, Validation, Visualization, Methodology, Writing - review and editing, Assisted with quantitative comparison bewteen BG505 SOSIP-displaying nanoparticles, Assisted with preparing the manuscript; Per Johan Klasse, Conceptualization, Formal analysis, Supervision, Validation, Visualization, Methodology, Writing - review and editing, Assisted with writing the paper, Assisted with quantitative comparison bewteen BG505 SOSIP-displaying nanoparticles; Andrew B Ward, Neil P King, Conceptualization, Formal analysis, Supervision, Funding acquisition, Validation, Visualization, Methodology, Writing - review and editing, Assisted with research design, Assisted with preparing the manuscript, Supervised the project; David Baker, Conceptualization, Resources, Software, Formal analysis, Supervision, Funding acquisition, Validation, Investigation, Visualization, Methodology, Writing - review and editing, Assisted with research design, Assisted with preparing the manuscript, Supervised the project

### Author ORCIDs
George Ueda (iD) https://orcid.org/0000-0002-9792-7149
Aleksandar Antanasijevic (iD) https://orcid.org/0000-0001-9452-8954
Matthew J Bick (iD) https://orcid.org/0000-0002-9585-859X
Philip JM Brouwer (iD) https://orcid.org/0000-0002-2902-7739
David Veesler (iD) https://orcid.org/0000-0002-6019-8675
Barney S Graham (iD) https://orcid.org/0000-0001-8112-0853
Per Johan Klasse (iD) https://orcid.org/0000-0001-8222-278X
Andrew B Ward (iD) http://orcid.org/0000-0001-7153-3769
David Baker (iD) https://orcid.org/0000-0001-7896-6217

### Decision letter and Author response
Decision letter https://doi.org/10.7554/eLife.57659.sa1
Author response https://doi.org/10.7554/eLife.57659.sa2

## Additional files

### Supplementary files
• Supplementary file 1. Sequences for all designed trimers, homo-oligomers, two-component nanoparticles, and antigen-fused components. (**A**) Amino acid sequences for all designed trimers and *de novo* homo-oligomers used for two-component nanoparticle design. Sequences include initiating methionines and His6-tags. Designed trimers that expressed solubly are denoted in bold, and experimental methods used for characterization are included in parentheses. *Components from previously described designed homo-oligomers in *Fallas et al., 2017* or the Protein Data Bank (PDB ID). (**B**) Amino acid sequences for all designed two-component nanoparticles. Sequences include initiating methionines and His6-tags. Designs that expressed solubly and co-eluted from IMAC are denoted in bold. Input oligomers from (**A**) are included in parentheses. (**C**) Amino acid sequences for all antigen-fused trimeric nanoparticle components. Sequences include initiating methionines and signal peptides.

• Transparent reporting form

### Data availability
Diffraction data have been deposited in the PDB under accession codes 6V8E and 6VEH. Cyro-EM structures have been deposited in the PDB under accession codes 6VFH, 6VFI, and 6VFJ. Electron density maps have been deposited in the EMDB with numbers 21162, 21163, 21164, 21165, 21166, 21167, 21168, 21169, 21170, 21171, 21172, 21173, and 21174. All data generated or analysed during this study are included in the manuscript and supporting files. Source data files have been provided for Figures 2, 4 and 6 specifically.

The following datasets were generated:

2000



| Author(s) | Year | Dataset title | Dataset URL | Database and Identifier |
|---|---|---|---|---|
| Sankaran B, Ueda G, Zwart PH, Baker D | 2020 | Tailored Design of Protein Nanoparticle Scaffolds for Viral Glycoprotein Immunogens | https://www.rcsb.org/structure/6V8E | RCSB Protein Data Bank, 6V8E |
| Bick MJ, Ueda G, Baker D | 2020 | Computationally designed C3-symmetric homotrimer from HEAT repeat protein | https://www.rcsb.org/structure/6VEH | RCSB Protein Data Bank, 6VEH |
| Antanasijevic A, Ueda G, Baker D, Ward AB | 2020 | De novo designed tetrahedral nanoparticle T33_dn2 | https://www.ebi.ac.uk/pdbe/entry/emdb/EMD-21162 | Electron Microscopy Data Bank, 21162 |
| Antanasijevic A, Ueda G, Baker D, Ward AB | 2020 | De novo designed tetrahedral nanoparticle T33_dn5 | https://www.ebi.ac.uk/pdbe/entry/emdb/EMD-21163 | Electron Microscopy Data Bank, 21163 |
| Antanasijevic A, Ueda G, Baker D, Ward AB | 2020 | De novo designed tetrahedral nanoparticle T33_dn10 | https://www.ebi.ac.uk/pdbe/entry/emdb/EMD-21164 | Electron Microscopy Data Bank, 21164 |
| Antanasijevic A, Ueda G, Baker D, Ward AB | 2020 | De novo designed octahedral nanoparticle O43_dn18 | https://www.ebi.ac.uk/pdbe/entry/emdb/EMD-21165 | Electron Microscopy Data Bank, 21165 |
| Antanasijevic A, Ueda G, Baker D, Ward AB | 2020 | De novo designed icosahedral nanoparticle I53_dn5 | https://www.ebi.ac.uk/pdbe/entry/emdb/EMD-21166 | Electron Microscopy Data Bank, 21166 |
| Antanasijevic A, Ueda G, Baker D, Ward AB | 2020 | BG505-SOSIP-T33_dn2A nanoparticle fusion component | https://www.ebi.ac.uk/pdbe/entry/emdb/EMD-21167 | Electron Microscopy Data Bank, 21167 |
| Antanasijevic A, Ueda G, Baker D, Ward AB | 2020 | BG505-SOSIP-T33_dn2A nanoparticle fusion component in complex with VRC01-Fab | https://www.ebi.ac.uk/pdbe/entry/emdb/EMD-21168 | Electron Microscopy Data Bank, 21168 |
| Antanasijevic A, Ueda G, Baker D, Ward AB | 2020 | De novo designed tetrahedral nanoparticle T33_dn2 presenting BG505-SOSIP | https://www.ebi.ac.uk/pdbe/entry/emdb/EMD-21169 | Electron Microscopy Data Bank, 21169 |
| Antanasijevic A, Ueda G, Baker D, Ward AB | 2020 | Tetrahedral nanoparticle T33_dn10 presenting BG505-SOSIP | https://www.ebi.ac.uk/pdbe/entry/emdb/EMD-21170 | Electron Microscopy Data Bank, 21170 |
| Antanasijevic A, Ueda G, Baker D, Ward AB | 2020 | Icosahedral Nanoparticle I53_dn5 presenting BG505-SOSIP | https://www.ebi.ac.uk/pdbe/entry/emdb/EMD-21171 | Electron Microscopy Data Bank, 21171 |
| Antanasijevic A, Ueda G, Baker D, Ward AB | 2020 | T33_dn10 | https://www.ebi.ac.uk/pdbe/entry/emdb/EMD-21172 | Electron Microscopy Data Bank, 21172 |
| Antanasijevic A, Ueda G, Ward AB, Baker D | 2020 | O43_dn18 | https://www.ebi.ac.uk/pdbe/entry/emdb/EMD-21173 | Electron Microscopy Data Bank, 21173 |
| Antanasijevic A, Ueda G, Baker D, Ward AB | 2020 | I53_dn5 | https://www.ebi.ac.uk/pdbe/entry/emdb/EMD-21174 | Electron Microscopy Data Bank, 21174 |
| Antanasijevic A, Ueda G, Baker D, Ward AB | 2020 | T33_dn10 | https://www.rcsb.org/structure/6VFH | RCSB Protein Data Bank, 6VFH |
| Antanasijevic A, Ueda G, Baker D, Ward AB | 2020 | O43_dn18 | https://www.rcsb.org/structure/6VFI | RCSB Protein Data Bank, 6VFI |
| Antanasijevic A, Ueda G, Baker D, Ward AB | 2020 | I53_dn5 | https://www.rcsb.org/structure/6VFJ | RCSB Protein Data Bank, 6VFJ |

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
