## [Decision Letter]

**Decision letter after peer review:**

Thank you for submitting your article "Tailored design of protein nanoparticle scaffolds for multivalent presentation of viral glycoprotein antigens" for consideration by *eLife*. Your article has been reviewed by José Faraldo-Gómez as the Senior Editor, a Reviewing Editor, and three reviewers. The reviewers have opted to remain anonymous.

The reviewers have discussed the reviews with one another and the Reviewing Editor has drafted this decision to help you prepare a revised submission.

Summary:

The manuscript describes computational nanoparticle design approaches, to create and optimize *de novo* scaffolds to display antigens such as HIV BG505 SOSIP, RSV DS-Cav1 and H1 HA. In case of SOSIP NPs, near-base epitopes availability to antibody binding was improved compared to previously published scaffold (I53-50NP, https://doi.org/10.1038/s41467-019-12080-1). Compared to I53-50NP, the new T33_dn2 NP displays the immunogen with a different valency.

The paper represents an evolution of the work of the same laboratories showing that antigens exposed on nanoparticles provide a superior immunogenicity. The approach is rigorous and the rationale is solid. However, in this paper, the authors do not ask the question of whether the new scaffolds improve the immunogenicity and therefore further experiments would be needed to prove comparable efficiency to induce potent B cell responses after immunization. Nevertheless, this paper stands on its own and could be published as it is if the paper is modified to address the reviewers' concerns, which predominantly center around the novelty of this paper compared with the authors' previous papers.

Essential revisions:

1) In this study the authors used tetrahedral, octahedral and icosahedral nanoparticles to display viral glycoproteins. EM and antibody binding experiments confirmed that the designed nanoparticles displayed intact "native like" HIV1 Env, Influenza hemagglutinin and perfusion RSV F trimers. It is unclear what the novelty is beyond decoration of nanoparticles with trimers. Nanoparticles of these symmetries were already reported by the PI previously. The methodology is not new. Decoration of said nanoparticles was also reported. Now they are decorating with viral trimers but they are not showing activity (beyond ELISA and SPR) and the decoration is suboptimal. The authors should clarify what is new in this paper.

2) Related concerns: Design of assemblies like these was already described; the attached antigens are not particularly rigid indicated by resolution; the Discussion section says they stabilized their fused antigens, but the data do not support this statement and there is no demonstration of biological activity. In particular, this statement: "This study demonstrates, both at the oligomer and nanoparticle design stages, the capability of generating protein based materials across a range of controllable sizes and topologies for targeted biological applications" is not new to this paper.

3) SPR experiments showed antibody accessibility on the tetrahedral scaffold was higher than for icosahedral scaffold, but the mAbs were immobilized. Would this still hold in solution?

4) Many of the EM micrographs are low quality – not many particles; particles broken; hardly any discernible shapes and reconstructions are noisy. CryoEM maps are low resolution even for nanoparticles alone. Assembles fusion even lower resolution and lots of density that was not interpreted

5) CryoEM of O43_dn18 1336 micrographs reported 103,356 particles, but the final reconstruction was out of 5050. So what are we looking at? How representative is this? Other reconstructions are 50k/140k and 30k/150k particles. Also, the FSC of T33_dn10 looks weird.

6) Reference models imposed appropriate symmetry, so the authors are selecting for the symmetry that matches. What % was that out of the particles that were picked?

7) The X-ray data were phased was phased by MR using the computational design as a search model, which could create bias.

8) What's a heavy atom RMSD? All atom? Subsection “Structural characterization of designed trimers”.

9) Why is *in vivo* versus *in vitro* assembly important?

10) Why are cysteines removed?

---

## [Author Response]

Essential revisions:1) In this study the authors used tetrahedral, octahedral and icosahedral nanoparticles to display viral glycoproteins. EM and antibody binding experiments confirmed that the designed nanoparticles displayed intact "native like" HIV1 Env, Influenza hemagglutinin and perfusion RSV F trimers. It is unclear what the novelty is beyond decoration of nanoparticles with trimers. Nanoparticles of these symmetries were already reported by the PI previously. The methodology is not new. Decoration of said nanoparticles was also reported. Now they are decorating with viral trimers but they are not showing activity (beyond ELISA and SPR) and the decoration is suboptimal. The authors should clarify what is new in this paper.

What is new in this paper is that we *de novo* designed nanoparticles from scratch specifically tailored for presenting viral glycoprotein antigens of interest. We have rewritten the first part of the manuscript to clarify this. In our previous work, we had designed nanoparticles using naturally occurring oligomers without vaccine applications in mind, and hence many of the previous nanoparticles were not compatible with viral glycoprotein display. Here, we first *de novo* design trimers with geometries tailored for fusion to specific viral glycoproteins, and then generate nanoparticles incorporating these designed trimers. This two-step design approach confers an added layer of geometric control, which was a key feature that allowed the glycoproteins to be fused to the designed nanoparticles without extremely long and flexible linkers. This high degree of geometric control and tunable antigen epitope accessibility offers a particularly well-suited class of molecules to probe B cell activation and the adaptive immune response, which was investigated in another study recently submitted to PLOS Pathogens (Carmen et al., 2020). In the current revised manuscript, we have made these points clearer.

2) Related concerns: Design of assemblies like these was already described; the attached antigens are not particularly rigid indicated by resolution; the Discussion section says they stabilized their fused antigens, but the data do not support this statement and there is no demonstration of biological activity. In particular, this statement: "This study demonstrates, both at the oligomer and nanoparticle design stages, the capability of generating protein based materials across a range of controllable sizes and topologies for targeted biological applications" is not new to this paper.

The novelty and significance of the new nanoparticles has been re-described throughout the main text, predominantly in the Introduction and Discussion section. A major goal of computational protein design is the generation of protein-based technologies that are tailored to specific applications. In our previous work generating nanoparticle immunogens, we essentially bolted antigens onto previously designed nanoparticles only when compatible. The advance reported in this work is that we have now generated novel self-assembling nanomaterials specifically tailored for genetic fusion to several trimeric viral glycoproteins and subsequent multivalent presentation. This represents a clear conceptual and technical advance over our previously published work which lacked this aspect of rational design. The revised text more directly conveys how these nanoparticles were generated from monomers with the added layer of geometric control and tunability at the oligomer design stage, representing the new accomplishment of targeted nanoparticle design for antigen presentation across various defined geometries.

Rigid presentation of antigens was not actively attempted in this study, although we were interested in characterizing this aspect of the antigen-displaying nanoparticles. Optimizing the termini positioning during trimer design allowed for short genetic linkers to be used in the antigen-displaying particles. Consequently, the fused antigens were readily discerned in 2D class-averages and reconstructed 3D classes, but they do not appear to be rigid, as the reviewer pointed out.

On the note of fused antigen stabilization, we now refer to a study submitted in parallel to PLOS Pathogens which includes data on melting temperatures and antigenicity of BG505 SOSIP fusions which supports our claims (Carmen et al., 2020). We also have previously observed increased thermal and antigenic stability of the DS-Cav1 antigen when fused to a naturally derived nanoparticle component (Marcandalli et al., 2119).

3) SPR experiments showed antibody accessibility on the tetrahedral scaffold was higher than for icosahedral scaffold, but the mAbs were immobilized. Would this still hold in solution?

Designed partly to simulate interactions with B cell receptors (note added to relevant Results section in main text), the immobilized mAb format exacerbates the steric hindrance of antigenic epitopes on the nanoparticles. In a previous study (Brouwer et al., 2019), we saw a significant degree of steric hindrance when icosahedral particles were immobilized and IgG mAbs flowed over. We have also seen only subtle reduction of mAb binding to immobilized tetrahedral particles (Carmen et al., 2020), which agrees with the format of immobilized mAbs in the present study. The immobilized mAb format shows greater reduction in access to antigenic epitopes on tetrahedral particles, but measurably less so than on icosahedral particles.

4) Many of the EM micrographs are low quality – not many particles; particles broken; hardly any discernible shapes and reconstructions are noisy. CryoEM maps are low resolution even for nanoparticles alone. Assembles fusion even lower resolution and lots of density that was not interpreted

To address the concern regarding the interpretability of EM data, we have now included higher quality NS-EM and cryo-EM micrographs and class averages in updated versions of Figure 3, Figure 4, Figure 5, and Figure 3—figure supplement 1.

Most of the described issues have to do with the flexibility within the designed nanoparticles and their stability. All antigens (HA, DS-Cav1, and BG505-SOSIP) are connected to the nanoparticle core via a short genetic linker that still allows for flexibility. Consequently, in the reconstructed EM maps, the local resolution for the antigen is significantly lower and the antigen-corresponding signal appears heterogeneous, with a lot of scattered density.

The nanoparticle scaffolds also have a certain degree of flexibility themselves. This is not observable at low resolution (i.e., in negative stain EM), but it becomes clearer with cryo-EM analysis. The best example is I53_dn5; in the 2D classes (Figure 3—figure supplement 1C) we observe 2D projection averages that are circular but also some that are closer to ellipsoid shape. The highest map resolution for this dataset was 5.35 Å, consistent with the existence of structural heterogeneity. Finally, some particles appear to be prone to disassembly under the EM conditions (i.e., during the preparation of NS grids). This is particularly the case with the T33_dn5 nanoparticle, as evidenced by the existence of partially assembled particles and free components in the raw micrographs in Figure 3B. Disassembly products are also visible in the BG505-SOSIP-presenting I53_dn5 nanoparticles (Figure 4C). Most likely, the disruption of internal cage structure is influenced by the removal of solvent during the blotting step and/or by the uranyl formate staining.

In the revised manuscript we have now included additional clarifications regarding the EM data to the Results section, further elaborating on the flexibility and stability of the designed nanoparticles.

5) CryoEM of O43_dn18 1336 micrographs reported 103,356 particles, but the final reconstruction was out of 5050. So what are we looking at? How representative is this? Other reconstructions are 50k/140k and 30k/150k particles. Also, the FSC of T33_dn10 looks weird.

2D and 3D classification steps are routinely applied to eliminate “bad” particles (i.e., contaminants and noise) from EM datasets, but also to select for the most homogeneous sub-population of “good” particles (i.e. particles corresponding to the imaged protein sample). This is performed to reconstruct a map with the most optimal combination of properties (high resolution, presence of relevant domains/features, map “smoothness” , interpretability, minimal fluctuations in local resolution, etc.) that can then be used to relax a molecular model. Along those lines, we have performed extensive 3D classification tests to recover the highest quality map for each of the tested nanoparticle samples; those maps have been reported here and submitted to the EMDB.

When we compared the resolution and quality of 3D maps from before and after the 3D classification step, we found only small improvements (see Table R1 for information on particle count and map resolution). The exception was I53_dn5, where an increase of ~0.7 Å in resolution was observed following 3D classification. We have not found any structural differences in maps before and after the 3D classification, and they are all in excellent agreement with the corresponding Rosetta-predicted models for each nanoparticle (Figure 3—figure supplement 1). Therefore, we believe that the reported maps are representative of the collected EM datasets and nanoparticles that were imaged. We have now updated Figure 3—figure supplement 1 with the resolution information for the intermediate 3D refinement steps.

The final FSC curve in the T33_dn10 datasets has a small drop at the intermediate spatial frequencies (corresponding to the ~4-6 Å resolution range). This behavior is independent of the mask tightness (data not shown) and appears to be originating from the reconstructed model itself. Variations in local resolution can influence the shape of the FSC curve. Nanoparticle components consist of a series of helix-loop-helix motifs. Local resolution is higher for the helices in the T33_dn10 map compared to the connecting loops, and this could have been the cause of the observed drop in FSC.

Table R1. Comparison of data processing stats before and after 3D classification step

* All reconstructions obtained in Relion 3.0 using 3D auto-refinement. Maps were then post-processed with the application of appropriate solvent mask (see Figure 3—figure supplement 1)

6) Reference models imposed appropriate symmetry, so the authors are selecting for the symmetry that matches. What % was that out of the particles that were picked?

This is an excellent question, but it is a very challenging one to answer. Due to the flexibility that exists within large complex nanoparticles on the level of individual components and/or at their interfaces, each nanoparticle can sample a certain conformational space. As a consequence, every single particle in an EM dataset is different from another particle and asymmetric on a certain level (or at a certain resolution). Map resolution is one of the parameters that can be used to at least partially characterize the consistency within each dataset. The fact that the T33_dn10 nanoparticle can be refined to 3.86 Å resolution indicates a higher level of consistency between individual particles in that dataset when compared to the I53_dn5 dataset, where the best resolution was 5.35 Å. This is further supported by the 2D classification results, where we observe the highest level of heterogeneity in the 2D class averages of I53_dn5 particles among the collected cryo-EM datasets (Figure 3—figure supplement 1). However, there are many other factors that need to be taken into consideration when comparing different EM data (e.g. microscope setup, aberration corrections, ice thickness, electron dose, radiation damage, etc.) so an exact comparison cannot be made.

7) The X-ray data were phased was phased by MR using the computational design as a search model, which could create bias.

As there existed no homologous structures, we used the computational design as a search model for MR. We recognize that all forms of MR introduce bias. Following MR, we used simulated annealing, ‘rebuild-in-place=false’ which uses the model for initial phasing but then wipes it and starts building from scratch, and ‘prime-and-switch’ phasing during Autobuild. These are all techniques designed to reduce model bias.

8) What's a heavy atom RMSD? All atom? Subsection “Structural characterization of designed trimers”.

Corrected to backbone r.m.s.d., and the values are included in Figure 2—figure supplement 3.

9) Why is in vivo versus *in vitro* assembly important?

Assembly in *E. coli* cells was achieved using bicistronic vectors encoding both designed nanoparticle components to screen for self-assembly by co-elution from IMAC purification (since only one component was His_6_-tagged). *In vitro* assembly enables production of each component independently, which is particularly useful for controlling the assembly of nanoparticles comprising fusion components. *In vitro* also allows for different expression host systems for each component. In this study for example, antigen-fused trimers were produced from HEK293F cells and partner components produced from *E. coli*, which is particularly relevant for large-scale manufacturing of vaccine candidates. There are currently three two-component nanoparticle vaccines in process development or cGMP manufacturing headed towards Phase I clinical trials using this strategy (Marcandalli et al., 2119; Boyoglu-Barnum et al., 2020).

10) Why are cysteines removed?

Cysteine removal was investigated to eliminate redox sensitivity in the protein, and to allow for future modification with new cysteines amenable to chemical conjugation.